# Demarcating geographic regions using community detection in commuting networks with significant self-loops

**Mark He[1], Joseph Glasser[1], Nathaniel Pritchard[2], Shankar Bhamidi[1], Nikhil Kaza[3]***

**1** Statistics & Operations Research, University of North Carolina at Chapel Hill, Chapel Hill, NC, United States of America, **2** Statistics, University of Wisconsin at Madison, Madison, WI, United States of America, **3** City & Regional Planning, University of North Carolina at Chapel Hill, Chapel Hill, NC, United States of America

* nkaza@unc.edu

**Data Availability Statement:** Data are available from: https://github.com/markhe1111/commuting-communities.

**Funding:** Mark He was funded by government support under contract FA9550-11-C-0028 and

## Abstract

We develop a method to identify statistically significant communities in a weighted network with a high proportion of self-looping weights. We use this method to find overlapping agglomerations of U.S. counties by representing inter-county commuting as a weighted network. We identify three types of communities; non-nodal, nodal and monads, which correspond to different types of regions. The results suggest that traditional regional delineations that rely on ad hoc thresholds do not account for important and pervasive connections that extend far beyond expected metropolitan boundaries or megaregions.

## 1 Introduction

Geographic regions map to social, cultural, and economic structures that enable us to make sense of the world. Demarcation of these regions allows institutional responses to shared problems by creating territorial administrations. These regions are useful at different scales and are created for varying purposes (e.g. cities, places, watersheds, economic regions) [1, 2, 3]. In the United States, metropolitan regions are conceived as collections of counties or equivalent areas (sub-state political units) and are used for different statistical, governance and planning purposes. Yet recent work suggests that these metropolitan regions have coalesced and that *megaregions* spanning multiple states to better project and plan for future growth [4]. At the same time, many urbanized areas are often sub-county regions [5].

Methods to identify these metropolitan areas, megaregions or labor markets are often subject to issues of scale and rely on ad hoc definitions, thresholds, and judgments about proximity, connections and similarity. For example, the U.S. Office of Management and Budget (OMB) uses commuting networks to define Core Based Statistical Areas (CBSA). Surrounding counties are added to a CBSA core county (population thresholds of 10,000 or 50,000 in urban areas for Micropolitan ($\mu$SA) and Metropolitan areas (MSA)), if social and economic ties (as defined by commuting to the core county) comprise more than 25% of the total ties. Adjacent CBSAs are merged into a single CBSA when the central county (or counties) of one CBSA qualifies as outlying with respect to the other CBSA. The county to CBSA relationship is a one-

awarded by the Department of Defense, Air Force Office of Scientific Research, National Defense Science and Engineering Graduate (NDSEG) Fellowship, 32 CFR 168a. Shankar Bhamidi was supported in part by NSF grants DMS-1613072, DMS-1606839 and ARO grant W911NF-17-1-0010.

**Competing interests:** The authors have declared that no competing interests exist.

to-one relationship; i.e. a county belongs to only one CBSA. If a county satisfies threshold requirements for multiple CBSAs, then it is allocated to the CBSA with which it has the strongest ties [6]. On the other hand, megaregions are defined by the Regional Planning Association with collections of counties that are part of CBSAs, then agglomerated using the Delphi method. Megaregions rarely overlap, but expert judgment is used to identify the boundaries when they do.

In both of these delineations, expert judgment is relied upon to set thresholds for the definition of core county, the strengths of connections, and where the boundaries fall. These judgments are colored by the analysts' orientation regarding the appropriate size of the region and the relevance of the connections. Judgments about boundaries are difficult to account for in analytic projects that seek to explain economic growth patterns [7], productivity gains [8] and city size distribution laws [9]. They are also particularly problematic when governmental assistance programs rely on statistical information conditioned by the boundaries (e.g. tax credits for investments in opportunity zones defined on the basis of Median Household Income in the MSA).

We present a method for inferring geographic regions systematically from the underlying data using community detection methods in network science. One of the key innovations of this approach is to identify multiple overlapping regions at different scales in the same statistical inference framework. We also extend the notion of community to identify nodal regions where peripheral connections are overwhelmed by connections to the core. We also extend community detection methods to include self-loops that have traditionally been implicit or ignored in other community detection work, but are of great importance in commuting networks. The results of these methods identify unusual regions that neither CBSA nor megaregions identify and allow a more nuanced approach to studying and governing metropolitan areas and labor markets [10]. In summary, our methods are able to differentiate between various types of communities that we classify into three major types:

1. **Monads**: Nodes preferentially attached to themselves in the sense that the self-looping proportions of these nodes are significantly stronger than the baseline self-looping proportions across the entire network.

2. **Nodal communities**: Peripheral nodes more strongly connected to some core nodes rather than among themselves, after accounting for the baseline self-loops.

3. **Non-nodal communities**: Clusters of nodes that are strongly connected to one another. (See Fig 1).

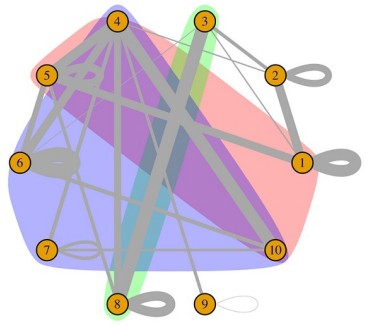 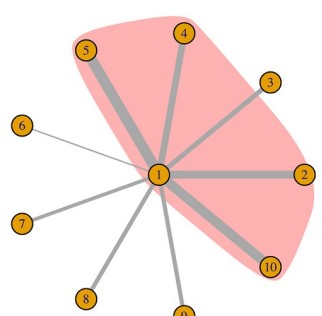 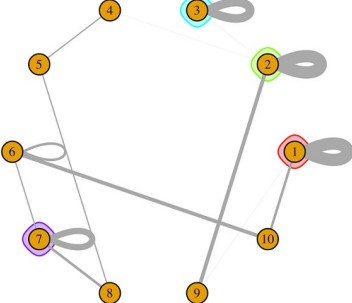

**Fig 1. Conceptual diagrams representing a) Overlapping non-nodal communities b) Nodal communities (trees) c) monads.** The different colors represent different clusters/regions.

Networks are used to model the relational structures between individual units of an observed system. A multitude of data structures may be conceived of as networks in the biological, physical, and social sciences. Over the last few years, owing to the explosion in data from a host of areas including social networks, information networks such as the Internet, and biochemical networks such as gene regulatory systems, there has been a concerted inter-disciplinary approach to understanding these data [11, 12, 13, 14, 15]. Due to the inherently relational nature of the commuting data, network methods offer a fitting approach for identifying clusters amongst interconnected regions.

## 1.1 Related work in region demarcation

Both CBSA and megaregions implicitly or explicitly rely on the notion of cores. In the case of the former, cores are counties. In the case of the latter, cores are CBSAs. The core-based approach to identifying urban regions has a history dating back to 1950s. Key to this approach is the identification of a core and its connections with the hinterland (nodal communities) [16]. This core-based approach ignores the emerging polycentric structure that has come to dominate regions around the world [17]. Since the 1980s, however, further studies have shown that most commuting flows in urban systems are lateral (i.e. between different parts of suburbs and hinterland) rather than core-centered [18]. Methods have been proposed to account for these peripheral commuting patterns and used to delineate regions [19]. In our approach, we eschew the a priori identification of cores and instead rely on entire commuting networks, thereby capturing important peripheral connections as well as polynodal and diffuse regions. We identify the core-centered regions in a posthoc analysis.

Non-unique membership is another problem that is not acknowledged in other approaches. Regional delineations tend to partition a set of geographic jfobjects instead of treating them as members in multiple agglomerations. Especially in densely urbanized regions, many counties have large numbers of commuters to different cities that are relatively close to one another [20, 21]. Often, delineations (such as OMB) tend to assign counties to one region by breaking membership ties rather than acknowledging connections with multiple regions. It is useful to relax the unique membership condition between an object and the agglomeration to which it belongs.

The other major issue that has received less attention in the literature, both in the context of regional demarcation as well as in the area of network science, is the idea of self-connection. Many agglomeration delineations in both network science and regional science have focused on the connection between two nodes/counties. However, commuting networks have significant self-loops (i.e. commuters within a county): 56% of the total commuters in 2010 in the US commuted within the county where they resided. Ignoring this large commuting pattern skews the results of agglomerations. Since some nodes may be preferentially attached to themselves (measured by the weight of the self-loop), they should be treated as their own agglomerations (see Section 4.4).

Community detection methods have been applied to commuting networks to identify regions but traditionally do not account for the above critiques. For example, Nelson and Rae ignore commuting within a node and focus only on commuting between nodes [22]. They also rely on a community detection algorithm that partitions the entire node set rather than identifying statistically significant connections and clusters. Our proposed approach identifies overlapping communities. Unlike their approach, which starts with census tracts, we start with counties because it is easier to fashion institutions for collections of political boundaries (counties) rather than statistical boundaries (census tracts).

## 1.2 Related work in community detection

Community detection is an approach used to divide a set of nodes in a given network into clusters whose nodes are strongly connected within a grouping as opposed to between groupings. Many techniques have been proposed for *unweighted* or binary data including modularity optimization [23, 24], stochastic block models [25, 26, 27, 28], and extraction methods [29, 30].

The field of community detection is vast, with a host of methodologies from many different fields including computer science, statistical physics, optimization and statistics. We refer the reader to the work of Fortunato [31] for a wide ranging survey. The two themes most closely related to the methodology developed in this paper are those that (1) establish a notion of quality functions such as modularity [32] and (2) statistical techniques for fitting empirical data to networks (such as fitting either Bayesian [33, 34, 35] or frequentist [36, 37, 38] models).

More precisely, one of the central techniques in community detection is modularity optimization [32], where for any assignment of community labels $\mathbf{c} = (c_1, c_2, \ldots, c_n)$ with $c_i \in \{1, 2, \ldots, K\}$ (if we assume there are at most $K$ communities) the modularity score of such a community assignment is

$$Q(\mathbf{c}) = \frac{1}{d_T} \sum_{u,v \in [n]} \left( A_{uv} - \frac{d_u d_v}{d_T} \right) \delta(c_i, c_j)$$

where $\delta$ is the Kronecker delta function with $\delta(x, y) = 1$ if $x = y$ and zero otherwise. A large number of community detection algorithms are built on (approximate) optimization of the modularity score.

Another class of methods is based on a network *null model*. Typically, in these schemes one constructs a network model that preserves some aspects of the observed network (in the context of unweighted networks). The preserved characteristic is often the degree distribution of the network (see Section 3.1 for details). Such a *scrambled network* with the same degree distribution creates a network with no inherent clustering tendency. The observed network is then compared with this null model to extract subsets which seem more densely connected within the subset as compared to the null model. We use the null-model based approach in our methodology (Section 3). More details on null models are further described in Section 3.1.

## 1.3 Related work in spatial network analysis

Substantial literature exists for the applications of spatial network analysis onto human mobility data. Barthelemy et al. [39] conducts a general survey of spatial network models and dynamic processes on these models. Batty et al. [40] provides a concise description of network methods specialized to the understanding of cities. Some popular network models for urban and spatial flows known as gravitation and radiation models are described in existing work [41, 42, 43].

Some studies use existing community detection techniques on novel geographical datasets to gain insight on how inferred communities are similar to or different from existing points of interest and how they change over time [44, 45]. Community detection methods on commuting behavior were used to partition Japan into synthetic cities in the work of Fujishima et al. [46]. Community detection techniques were used to understand the polycentric spatial patterns within Singapore [47]. Other work uses proxy measures for mobility patterns, such as the flow of banknotes through the US [48], mobile phone data [49] and vehicle GPS tracking data [50], then applies existing community detection techniques. For a comprehensive bibliography of models utilizing network analytic techniques related to human mobility flow and their

associated data, see the work of Pappalardo et al. [51]. Conceptually, our work differs from most of the existing work as one of the primary motivations is developing new community detection techniques specifically tied to commuting networks that have a high proportion of self-loop weights.

The rest of the paper proceeds as follows. We first briefly describe the commuting dataset for the conterminous United States in 2010. We then present a method to identify clusters in a weighted network with self-loops. Using this method, we analyze the commuting patterns between counties in the United States in 2010 to identify the clusters of counties that are significantly connected. We then present and discuss the results of regional delineation and compare them with other delineations. We conclude with the limitations of our analysis and potential future research directions.

## 2 Data description

We downloaded our data from the US Census Bureau's Local Origin Destination Employment Statistics (LODES). This dataset contains commuter data between census tracts for all of the continental United States in the year 2010, which we then aggregated to the county level. The data are stored as an undirected and weighted network with self-loops such that each node represents a county and the weight on each edge represents the number of commuters between the connected counties. Edges with fewer than 100 commuters are removed from the network. Commuters who travel more than 100km (distance between population weighted centroids) are also ignored to remove the effect of telecommuters or super commuters similar to [22]. The resulting network contains 3,091 nodes and 17,632 edges. Los Angeles County has the largest number of commuters to itself ($\sim$ 3.1 million), while Los Angeles County to Orange County in California is the largest non self-loop edge ($\sim$ 0.6 million). As stated earlier, about 56% of the commuters are commuting within the county. As such, this network can be described as a **strongly self-looping network**.

## 3 Configuration null model

The Configuration Model, first introduced by Bollobas and Bender [52, 53], is a probability measure on a family of multigraphs that preserves the degree sequence. The input to the model is an observed graph from which we extract the degree sequence, namely a list consisting of the vertices and their corresponding degrees. The model then constructs a *random graph* as follows: start with the degrees of nodes with $d_u$ denoting the degree of vertex $u$; associate every vertex $u$ with $d_u$ "stubs". One then performs a uniform matching on these stubs to form full edges, thus resulting in a random graph with the prescribed degree sequence but without any other inherent clustering tendency. The relative proclivity of each node to form ties is determined purely upon its degree.

Many of the aforementioned community detection methods utilize the configuration model as the null model [30, 54, 55, 32]. We significantly extend the methodology developed by Palowitch et al. [56] for weighted network data by developing a new null model for **weighted networks with self-loops**. The outcome of the methodology reveals both significantly connected communities, monads, as well as nodal communities in the context of regional commuting flows (see Fig 1).

### 3.1 Related null models

A null model in the context of community detection is a random network model which preserves some aspects of an observed network but without any explicit community structure. The most common null model used in the context of unweighted networks is the configuration

model (described in Section 1.2). Once a null model for the network is established, communities based on the null model are groups of nodes that deviate from the baseline by being more connected to each other than expected under the null.

Various functionals are used to measure the deviation of a set or a partition of the entire node set from the null model. The most popular among these functionals is modularity score [23, 24]. One can then try to optimize such scores to find the best partitions. Fosdick et al. introduced a framework for configuration models that accounts for self loops and used a modularity optimization approach for community detection as an application, but did not focus on weighted networks whose self loops account for the majority of its weights [54]. Another null-model based approach is to assess the statistical significance of the deviation of subsets from what one would expect under the null, correcting these estimates for false-discovery rates, and then extracting communities that appear to be more significantly connected than under the null model. Several approaches have implicitly utilized the notion of deviation against the null partitions, such as likelihood ratios [28] or Bayes factors [27].

Though other community detection methods often allow self loops, our method explicitly uses a parameter to account for their effects and integrates them in a iterative testing framework. Most existing methods that allow for self loops, or presume that self-loops are somehow *similar* in characteristic to the other edges, do not properly account for when self loops are large, as in the case of the algorithm introduced by Palowitch et al. [56]. Previous tree-based methods do mention self-loops, but few focus explicitly on strongly self looping networks, where self loops account for over 50% of the weights. Xiang et al. introduced one such tree-based method [57], which uses a modularity measure that is rescaled by the size of the self loop. Peixoto's allows for self loops in the Bayesian stochastic blockmodel formulation [35]. The method of Cafieri et al. also accounts for self loops in the context of modularity maximization [58].

## 3.2 Notation

We denote an undirected weighted network on $n$ nodes by the triple $G = ([n], \mathbf{A}, \mathbf{W})$, where $[n] := \{1, 2, \ldots, n\}$ is the set of $n$ labeled nodes; $\mathbf{A} = (A_{uv})$ is an $n \times n$ square, symmetric adjacency matrix with $A_{uv} = 1$ if there is an edge between $u$ and $v$, and $A_{uv} = 0$ otherwise. Since we are interested in networks with self-loops, we assume $A_{uu} \equiv 1$ for all $u \in [n]$. Though conventionally the self-loop edge is defined as $A_{uu} = 2$, we define it to be 1 as this convention makes the algebra simple when defining the null model. We let $\mathbf{W} = (W_{uv})$ be another symmetric matrix representing (non-negative) weights on edges with $W_{uv}$ denoting the weight between $u$, $v \in [n]$ with $W_{uv} \equiv 0$ if there is no edge between $u$ and $v$. We let $d_u = \sum_{v \in [n], v \neq u} A_{uv}$ denote the degree of a vertex, which specifically is the total number of edges connecting to $u$ ignoring self-loops. The **total** strength of a node $u$ is defined as: $s_u = \sum_{v \in [n]} W_{uv}$. We let $d_T = \sum_{u \in [n]} d_u$ and $s_T = \sum_{u \in [n]} s_u$ denote the total degree and weight of the network, respectively. We define $\varrho_u$ to be the propensity of the node to connect to itself by

$$\varrho_u := \frac{W_{uu}}{s_u}, \qquad u \in [n].$$

We define the baseline propensity of the self-loop ratio for the entire network to be:

$$p = \frac{\sum_{u \in [n]} W_{uu}}{s_T}. \tag{1}$$

We let $\mathbf{d} = (d_1, \ldots, d_n)$ and $\mathbf{s} = (s_1, \ldots, s_n)$ denote the degrees and strengths of nodes in $[n]$, respectively.

### 3.3 Continuous configuration model extraction

Significance-based testing was directly pursued in the context of unweighted networks in [59] and weighted networks in [56]. We extend the significance testing based approach developed by Palowitch et al. [56] in scope and application by adjusting for self-loops. Palowitch et al. [56] developed a method that used a weighted configuration model as a null model that preserved the expected degrees and strengths of any given node $u$ with its actual respective strengths and degrees. The assumptions of the configuration model are as follows:

$$\mathbb{E}(D(u)) = d_u, \qquad \mathbb{E}(S(u)) = s_u \tag{2}$$

We refer to this method as *CCME*.

**3.3.1 Model construction.**  We start by describing the null model for weighted networks with self-loops that will serve as a comparative model for an observed weighted network $G = ([n], \mathbf{A}, \mathbf{W})$. The model is indexed by a family of parameters $\boldsymbol{\theta} = (\mathbf{d}, \mathbf{s}, \kappa_{SL}, \kappa_{nSL}, a, b)$ where $\mathbf{d}, \mathbf{s}$ are, as before, the degree and weight sequences of the observed network, respectively, $\kappa_{SL}, \kappa_{nSL} > 0$ are parameters that control the variance of self-loop and non self-loop edge distribution in the null model and $a, b > 0$ are parameters constrained by the relation $a/(a + b) = p$ where $p$ is, as in (1), the global self-looping tendency of the observed graph. The concentration parameters $a, b$ of the beta distribution with mean $p$ represent the sparseness and tail shapes (tendencies towards zero or one) of the self-looping probability.

Implicitly, we fix two distributions $F_{SL}$ and $F_{nSL}$ on $\mathbb{R}_+$ with **mean one** and variance $\kappa_{SL}$ and $\kappa_{nSL}$ respectively. Using the above ingredients we construct a random weighted graph $\mathcal{G} = ([n], \hat{\mathbf{A}}, \hat{\mathbf{W}})$ as follows:

(i).  **Network topology**: By design $\hat{A}_{uu} = 1$ for all $u \in [n]$. For all $u \neq v$ we let

$$\mathbb{P}(\hat{A}_{uv} = 1) = \frac{d_u d_v}{d_T}. \tag{3}$$

(ii).  **Self-loop edges**: For self-loop edges, we generate edge strengths as follows: First for each vertex $u \in [n]$ (independently across vertices), we generate its *self-loop propensity* $\hat{\varrho}_u \sim \text{Beta}(a, b)$ (i.e. a Beta distribution). Next we generate $\xi_{uu}$ from distribution $F_{SL}$ (independent of $\hat{\varrho}_u$). Then, we model

$$\hat{W}_{uu} := \hat{\varrho}_u s_u \xi_{uu}. \tag{4}$$

(iii).  **Non self-loop edges**: For $u \neq v$ generate edge strengths as follows: first if $\hat{A}_{uv} = 0$ from step (i) then let $\hat{W}_{uv} = 0$. If $\hat{A}_{uv} = 1$ then let $\xi_{uv} \sim F_{nSL}$, and let

$$\hat{W}_{uv} = (1 - \hat{\varrho}_u) q_{uv} \xi_{uv}. \tag{5}$$

where each $q_{uv}$ represents the following ratio of strengths and degrees of $u$ and $v$:

$$q_{uv} = \frac{s_u s_v}{s_T} \bigg/ \frac{d_u d_v}{d_T}, \tag{6}$$

Writing $D(u)$ and $S(u)$ for the degree and strength of vertex $u$ in the associated random graph, it is easy to check that

$$\mathbb{E}(D(u)) = d_u, \qquad \mathbb{E}(S(u)) = s_u, \qquad \mathbb{E}(\hat{W}_{uu}) = ps_u. \tag{7}$$

The weight matrix and adjacency matrix represent inherently different, though correlated, modes of relation. For example, in a social network one can imagine two individuals having similar degrees but very different rates of interaction with the individuals they are connected to. In the context of the commuting data, Mesa County, CO and L.A. County, CA have similar degree but very different strengths. Thus part of the aim of this paper was to develop a baseline null model that would preserve both degrees and strengths as well as a baseline level of self-loopiness and then compare an empirically observed network against this null model to extract regions of significantly higher connectivity after accounting for this baseline connectivity.

We note that in (7), the first two conditions are identical to that of the 'ordinary' CCME method, but the third which preserves the ratios of expected self-loops is novel. The model preserves (on average) the degrees and strengths of the observed graph without any other specific notion of clustering. Each vertex has no particular preferential self-looping proclivity other then the average tendency $p$ of the entire network. We refer to this model as CCME with self-loop adjustment (*CCME-SL*).

### 3.4 Parameter specifications

Palowitch et.al [56] use a method-of-moments estimator to specify parameters for CCME. We use this method to *learn* the parameters from the observed graph. We specify two types of variables to describe the uncertainty arising out of the strengths of the nodes' connection propensities.

Recall that we denote $\kappa_{SL}$ as the variance of the self-looping edge weight distribution (with distribution $F_{SL}$) and $\kappa_{nSL}$ as the variance of the non-self-looping edge weight distribution (with distribution $F_{nSL}$). Both of these variables have mean one to ensure the preservation of the strengths and degrees for the configuration null model and to ensure identifiability.

**3.4.1 Variance parameters.** The method-of-moments estimates for $\kappa_{SL}$ and $\kappa_{nSL}$ are as follows:

$$\hat{\kappa}_{SL} = \frac{\sum_{u \in [n]} (W_{uu} - ps_u)^2 - \sum_{u \in [n]} s_u^2 \hat{\sigma}_p^2}{\sum_{u \in [n]} s_u^2 (\hat{\sigma}_p^2 + p^2)} \tag{8}$$

$$\hat{\kappa}_{nSL} = \frac{\sum_{u \in [n]} \sum_{v \neq u} (W_{uv} - (1-p)q_{uv})^2 - \hat{\sigma}_p^2 \sum_{u \in [n]} \sum_{v \neq u} q_{uv}^2}{(\hat{\sigma}_p^2 + (1-p)^2) \sum_{u \in [n]} \sum_{v \neq u} q_{uv}^2}, \tag{9}$$

where $\hat{\sigma}_p^2$ represents the estimated variance of $\varrho_u$ using empirical method-of-moments, $q_{uv}$ represents the ratio of strengths to degrees as described in (6).

$$\hat{\sigma}_p^2 := \mathrm{Var}(\hat{\varrho}_u) = \frac{1}{n-1} \sum_{u \in [n]} \left( \frac{W_{uu}}{s_u} - p \right)^2. \tag{10}$$

$\hat{\kappa}_{nSL}$ represents the variation in relative weights between two edges when we know that the strengths (total sum of weights) are the same. $\hat{\kappa}_{SL}$ represents the variation within a single self-directed edge. These estimates account for the inherent variability of the edge weights of an empirically observed network. The eventual proposed score function (Section 4.2) used to judge the significance of the internal connectivity structure of a community (or any subset of

nodes) can use this variability metric in its calibration of significance. Details on the derivations of these parameters can be found in S1 Appendix A.

**3.4.2 Beta random variable to model self looping proportion.** We specify $\varrho_u$ as adhering to a Beta($a$, $b$) distribution independent across $u \in [n]$ with mean $p$ and with variance equal to $\hat{\sigma}_p^2$, the sample variance of $\{\varrho_u : u \in [n]\}$. The beta distribution is supported on [0, 1]. We designate the proportion of self-looping commuters in each node as comprised of the averages of decisions to either commute *in-county* or *out-of-county* by a host of commuters. The empirical distribution of $\hat{\varrho}_u$ closely matches the simulated values of $\varrho_u$, except for a few nodes that are at the upper or lower end of the distribution (in S1 Appendix). We note that for a beta distribution,

$$\mathbb{E}(\varrho_u) = \frac{a}{a+b} = p; \quad \text{Var}(\varrho_u) = \frac{ab}{(a+b)^2(a+b+1)} := \sigma_p^2 \tag{11}$$

We use $p$ and $\hat{\sigma}_p^2$ (10) to determine $a$ and $b$ and, from (11), express their estimates as

$$\hat{a} = \frac{-p(p^2 - p + \hat{\sigma}_p^2)}{\hat{\sigma}_p^2}; \quad \hat{b} = \frac{(p-1)(p^2 - p + \hat{\sigma}_p^2)}{\hat{\sigma}_p^2}. \tag{12}$$

# 4 Community detection algorithm

The CCME-SL algorithm is split into three general phases: *initialization, update*, and *filtering*. These steps compose the general procedure of iterative testing. Significant communities are groups of nodes with cross-edges that deviate considerably from the expected values under a null model. Significant communities are determined by repeatedly applying an iterative search algorithm that starts with a seed set $B_0$ and finds all nodes with edges connecting to the seed set. The edge-weights are then summed as a test statistic which is evaluated against the expected values of the sums of the weights in the set under the null model (described in S1 Appendix A.4) with respect to each node in the starting seed set $B_0$, imputing a p-value for each node.

Each p-value from the candidate set is rejected if it is significant after being corrected by the Benjamini-Hochberg correction. The nodes with p-values that are *significant* in the present iteration are used as the initial seed sets for the next iteration. The final set $B$ is extracted when the node-set becomes stable: when at some iteration step $k$, $B_k = B_{k+1}$. Nodes in the final set have a stronger affiliation with each other and have fewer edge connections with all other nodes outside the set.

In this section, we describe each of the phases of CCME-SL in detail. We also describe the hub and monad detection steps as post-community detection phases of the method.

## 4.1 Initialization

We initialize (step 1) sets $B_0$ by setting counties with high commuting volume as seed nodes (which represent counties). We select nodes that have above 20,000 self-commuters as seed nodes. We select these nodes because they are proxies for relative population centers where commuter traffic radiates outwards to more peripheral connections upon each iteration. We then find all nodes which are connected to each seed node. The seed node and its connected nodes are used as the initializing sets $B_0$. The seeds are largely irrelevant to the final outcome: the final outcomes reveal similar outcomes regardless of what the initial nodes selected are, so long as a majority of the high-volume nodes are included (see Fig 4). However, because the

initial seed nodes are fixed using the above heuristic, the algorithm converges to the same resulting clusters under the same parameters $\alpha$ and $\tau$.

## 4.2 Update

Stable communities are found using an iterative node-set updating scheme based on the p-value of the connectivity between a single node $u \in [n]$ and a candidate (testing) set $B \in [n]$. We denote $S(u, B, G)$ as the *connectivity* of a single node to the set of nodes which is hypothesized to be a community:

$$S(u, B, G) = \sum_{v \in B} W_{uv}.$$

When the observed value $S(u, B, G)$ significantly exceeds the expected sum of weights under the null model, then there is evidence to support the claim that there is some additional structure undergirding the set of nodes than that which is posited by the null model. The null model attributes connectivity between sets of nodes as dependent only on the strengths and degrees of the aggregations of the nodes themselves.

The p-value representing the significance of a node-set is given by:

$$p(u, B, \mathcal{G}) = \mathbb{P}(S(u, B, G) > S(u, B, \mathcal{G})).$$

In the above equation $G$ is observed but $\mathcal{G}$ is random with a distribution given by the null model $\mathbb{P}$ and with each $B$ representing the candidate set to be tested. When the observed value of $S(u, B, \mathcal{G})$ is much larger than the expected value, expressed as $S(u, B, G)$, the p-value is low. Low p-values are rejected in an iterative fashion so as to allow the formation of node-sets with edges that are consistently significantly connected to each other. We define these sets as communities.

The iterative method is described as follows: for each $u \in [n]$, given a set $B$ (or denoted by $B_k$ at $k^{th}$ iteration), we find all counties that are connected to the present set, then p-values are imputed for members of the candidate set $B$ and repeatedly tested until the set becomes stable upon sequential iterations:

1. Calculate p-values $\mathbf{p} = p(u, B, \mathcal{G})$. P-values are calculated using a normal approximation for the distribution of $S(u, B, \mathcal{G})$. Details on this part of the procedure are given in S1 Appendix A.4

2. Obtain threshold $\tau(\mathbf{p})$ using a Benjamini-Hochberg multiple testing procedure [60]. The procedure is used for sets of p-values that are obtained through multiple hypothesis testing. The rejection method ensures that the expected number of falsely rejected hypotheses divided by the total number of rejected hypotheses (false discovery rate, or FDR) has a maximum percentage of $\alpha$. A false discovery rate threshold $\alpha$ of 0.05 is common in many applications, but for community detection we find empirically that such a threshold should be lower to avoid excess overlaps.

3. The next set reached by the iteration is defined as $B' = \{u : p(u, B, \mathcal{G}) \leq \tau(\mathbf{p})\}$

The above steps are iterated with $B'$ replacing $B$ until we reach a fixed point. We set $\alpha = 0.01$ at each step of iterative testing to perform community detection. The threshold can be made higher or lower, and such adjustments do not change the results drastically (see Supporting Information for details), but the threshold of .01 appears to be optimal for maximizing coverage and minimizing overlaps.

## 4.3 Filtering

After obtaining $M$ stable communities $C^j$, where $j = 1, \ldots, M$, we remove redundant node-sets that have a high proportion of overlap with other sets. Redundancy in clusters is evaluated using the Jaccard similarity index. A Jaccard similarity index of two sets is defined as the ratio of the size of common elements between the two sets over the total distinct elements, or concisely expressed as $J(A, B) = |A \bigcap B|/|A \bigcup B|$ for two sets $A$, $B$ [61].

We evaluate Jaccard similarities for each pair of found communities $C^i$, $C^j$. If the Jaccard index $J(C^i, C^j)$ is above a given pre-set threshold $\tau$, then the clusters are redundant and we select a preferred cluster by calculating the average weight per connection between nodes. We use a simple formula for a given stable node-set $C$:

$$K(C) = \frac{\sum_{v \in C, v \neq u} \sum_{u \in C} W_{uv}}{|C|}.$$

$K(C)$ roughly measures the average sum of weights among cross-edges per node within a candidate set $C$. Given that two sets $C^i$, $C^j$ have Jaccard overlaps larger than $\tau$, a higher $K(C^i)$ compared to $K(C^j)$ signifies more interconnectivity between nodes in $C^i$ and thus it is kept in the final set of communities while $C^j$ is removed. We set the $\tau$ parameter to be 0.80 when implementing the method on commuting networks.

## 4.4 Detection of monads

One unique feature of geographical commuting networks is that a non-trivial proportion of the total commuting volume is not found in edges across vertices because most residents commutes within their counties. We define the degree of *monadicity* of a given node as the following:

$$I_u := W_{uu} - ps_u.$$

Assuming the observed graph originated from the null model, the variable $I_u$ measures two things. Firstly, $I_u$ measures how much larger $\varrho_u$ is for a given $u$ than the global mean self-looping tendency $p$. Secondly, $I_u$ measures how much larger the latent $\xi_{uu}$ component of $\hat{W}_{uu}$ is than its expected value of one (recall that self-loop weights are modeled as $\hat{W}_{uu} = \hat{\varrho}_u \xi_{uu} s_u$ from (4)).

The exact form of the variance of $I_u$ is difficult to calculate, but we use a simulation method to approximate a p-value for $I_u$. We first determine estimates for $a$, $b$ from the mean and variance of $\varrho_u$ under the null model. Given $p$ and the sample standard deviation $\hat{\sigma}_p^2$, we find estimates for $a$, $b$ from (12). We use these estimated parameters to simulate a measurement of how extreme the $\hat{I}_u$ of a given node is, compared to that of a measurement assuming random generation from a beta$(\hat{a}, \hat{b})$ distribution. If the the actual value $I_u$ is large, as measured by whether it is above the $\alpha$-th quantile of the simulated values, then the node is deemed significant because it consistently exceeds what would be expected if $\varrho_u$ were randomly generated from a distribution of the same parameters. Such a simulation-based method to approximate p-values is commonly used [62].

The process of finding significant *monads* may be concisely described by the following procedures. First, we simulate $\tilde{\varrho}_u$ from beta$(\hat{a}, \hat{b})$ for every node (county). We then obtain the empirical distributions of how monadic a given node is by computing the empirical distribution of $\hat{I}_u = W_{uu} - \tilde{\varrho}_u s_u$. Following this step, we consider $I_u = W_{uu} - ps_u$. If $I_u$ is in the $1 - \alpha^{\text{th}}$ tail of the distribution $\hat{I}_u$ then the node is monadic at this instance of simulation. We repeat

the procedure 10,000 times and the nodes that are monadic all of the 10,000 trials are conclusively classified as monads. In practice, we set $\alpha$ equal to .05 for this test. The resultant group of nodes are significantly monadic at the 5% significance level.

## 4.5 Differentiating nodal communities from non-nodal communities

In the post-processing phase, we identify nodal communities within the communities detected by using the local clustering coefficient as defined by Opshal et al. [63] for weighted networks. For an unweighted network, the local clustering coefficient of a node $u$ is the ratio of the number of present ties over the total number of possible ties between the node's neighbors. A community will have a low clustering coefficient if there is a single hub-like node that is connected to all its peripheral nodes, but its peripheral nodes do not connect to each other. A node in a complete graph has a coefficient of 1.

For a weighted network, Opsahl et al. define the minimum clustering coefficient of a particular node $u$ in a set $C$ using *triplets* and *triangles* of nodes [63]. A triplet $\delta(u, v, w)$ is defined as a set of three nodes that share at least one edge with another node in the set. A closed triangle $\Delta(u, v, w)$ is a set of three nodes whose nodes all connect to *both* other nodes in the set. One may visually conceive of a triplet as a loosely connected set of three vertices which may have a missing edge, and a closed triangle as a clique of three vertices with three edges.

Using triplets and closed triangles, Opsahl et al. [63] define the minimum clustering coefficient of node $u$ in the set $C$ as

$$m(u, C) = \sum_{v,y:\Delta(u,v,y)\in C} \min\left(W_{uv}, W_{vy}\right) \Bigg/ \sum_{v,y:\delta(u,v,y)\in C} \min\left(W_{uv}, W_{vy}\right).$$

The ratio $m(u, C)$ is the ratio of the sum of all closed triangles to triplets associated with $u$. If the sum of the minimum values of triangles is how compared to those of all triplets within a cluster, then that implies the presence of a dominant node that projects high edge weights across many peripheral nodes. A low clustering coefficient signifies that communities are bound together by a common node, while a high $m(u, C)$ signifies that the nodes are all connected to each other.

We determine the overall clustering coefficient of a community by

$$m_{\text{total}}(C) = \sum_{v \in C} \frac{(1 - m(v, C))s_v(C)}{\sum_{u \in C} s_u(C)}. \tag{13}$$

$m_{\text{total}}(C)$ is constructed to capture how tree-like the highest-weight nodes are in a given cluster $C$. The lower $m(u, C)$ is, the more tree-like the node is. $1 - m(u, C)$ weighted by the strengths of nodes in a given cluster assigns a value of how tree-like and strong a node is. Summing these values gives an overall measure of the monocentricity of a cluster, as the strongest nodes tend to have the smallest $m(u, C)$. The empirical distribution of $m_{\text{total}}(C)$ is bimodal (see Fig 2) with a split around 0.4. We use this value to identify nodal communities.

## 4.6 Methods to compare communities with other delineations

We compare our results with other existing delineations (in particular OMB's Metropolitan areas) by using the Fuzzy Rand Index (FRI) [64]. The Fuzzy Rand Index is a metric that measures the similarity of two covers, $C_1$ and $C_2$. A cover $C$ is the assignment of vertices of a graph into $k$ groups, where a vertex may belong to more than one group. Counties that are not assigned to any community will belong to their own group, a group of counties that do not belong to any community. Let $V$ represent the set of vertices and $C$ represent a cover of $V$.

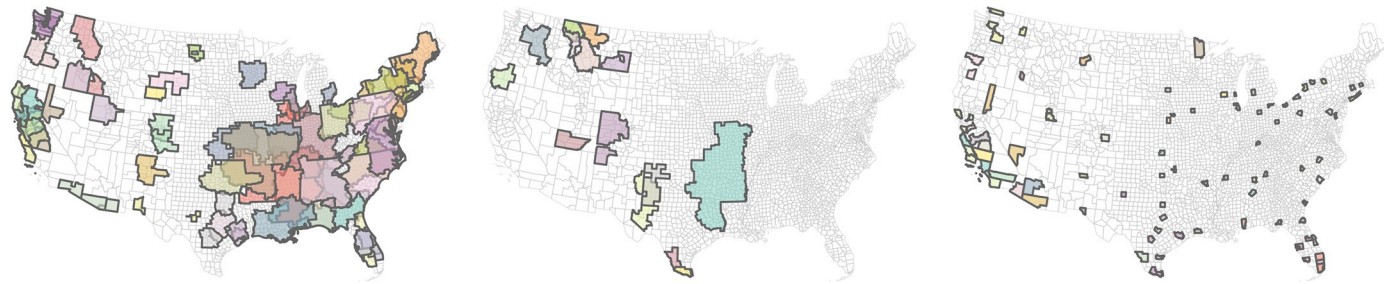

**Fig 2. Resulting communities from the CCME-SL algorithm.** Communities (non-nodal) (left), nodal clusters (middle), and monads (right).

Each element $v \in V$ is characterized by its membership vector, $C(v)$, which describes the degree of membership between a vertex and each group. A membership vector is subject to the following constraints: $C(v) = \{C_1(v), C_2(v), \ldots, C_k(v)\} \in [0, 1]^k$, where $C_i(v)$ is the degree of membership of $v$ in the $i^{th}$ community, $C_i$, and $\sum_{i \in 1, \ldots, k} C_i(u) = 1$. The norm of $C(u) - C(v)$ in a cover with $k$ communities will be defined to be $\|C(u) - C(v)\| = \sum_{i \in 1, \ldots, k} |C_i(u) - C_i(v)|/2$. For some cover $C$, the similarity measure between two nodes $u$, $v$ is defined as $E_C(u, v) = 1 - \|C(u) - C(v)\|$. The distance measure between two covers, $C_1$ and $C_2$ of a network $V$ is defined as:

$$d(C_1, C_2) = \frac{\sum_{u,v \in V} |E_{C_1}(u, v) - E_{C_2}(u, v)|}{k(k-1)/2}.$$

Likewise, the similarity measure between two covers is $1 - d(C_1, C_2)$.

## 5 Results

We use the term 'clusters' to refer to all sets of counties that we obtain from the detection procedures, 'communities' to refer to clusters that contain more than one county, and 'nodal communities' to refer to clusters that have strong nodal centers with little lateral commuting. We use the term 'monads' to refer to those counties that are strongly connected to themselves.

From the total 3,091 US counties, we find a total of 182 significant clusters. Of these clusters, 14 are nodal communities, 78 are non-nodal communities, and 90 are monads. Together they cover 90.3% of the population of commuters (93% intra-county, 87% inter-county). The method simultaneously delineates both small (such as monads and dyads) and large clusters consisting of hundreds of counties (see Table 1). For example, Santa Barbara and San Luis Obispo counties in California are strongly connected to one another and are separate from

**Table 1. Size characteristics of the identified regions.** IQR gives the first and third quartiles of the cluster sizes. Note that the number of counties is the total number captured by these regions with duplicates removed to avoid double counting within each subgroup.

| Cluster Type | Count | Mean | Median | IQR | Min | Max | # Counties |
|---|---|---|---|---|---|---|---|
| Non-nodal communities | 78 | 49 | 15 | [8, 46] | 2 | 356 | 2044 |
| Nodal communities | 14 | 28 | 5.5 | [4, 19] | 4 | 255 | 392 |
| Monads | 90 | 1 | 1 | | 1 | 1 | 90 |
| Comparison Delineations | | | | | | | |
| Metropolitan Statistical Areas (MSAs) | 363 | 3.0 | 2 | [1, 3] | 1 | 28 | 1094 |
| Megaregions | 12 | 80 | 38 | [27, 96] | 8 | 388 | 967 |

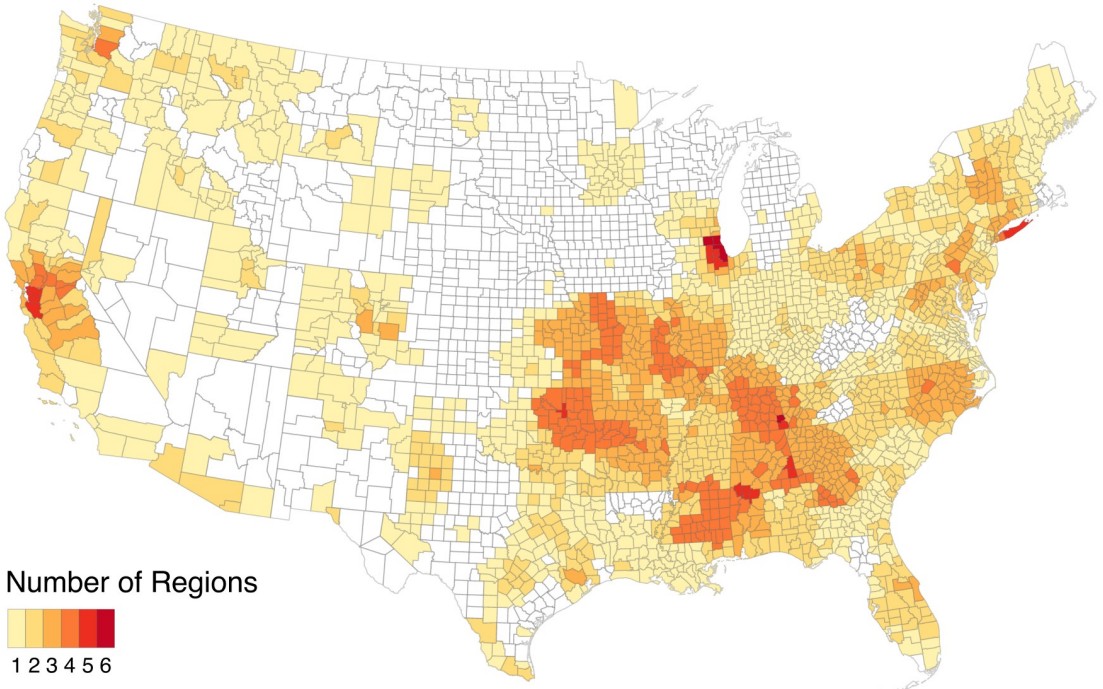

**Fig 3. Heatmap of frequencies of each county to appear in any cluster (community, nodal cluster, or monad).**

other clusters. Of the regions identified, 99 are monadic or dyadic counties. 68 of these clusters are medium sized, comprising between 2 and 50 counties. In 20 other instances, CCME-SL identifies clusters consisting of 50 or more counties that span several states. These tend to be polycentric regions centered around multiple large cities such as Philadelphia, Washington DC, and Baltimore (see Fig 5). Out of the 182 total clusters, the average size of a cluster is 24 counties, with a standard deviation of 59. The median size, however, is only 2, signifying that the majority of imputed clusters are monads. When only considering communities, the average size is 49, with a standard deviation of 78. The median size of communities is 15, suggesting that the distribution of community sizes is right-skewed (see Table 1). In general, many of the counties belong to overlapping clusters, with some belonging to as many as six different clusters (see Fig 3).

The parameters associated with the null model are as follows: $p$ is.46. $p$ is roughly the mean of intra-county commuters; though weights between cross-edges are counted twice, the proportion of intra-county commuters is 56%. $\hat{\sigma}_p^2$, the sample standard deviation of $\varrho_u$, is.04, signifying that around half of the commuting volume in the US does not leave the county where it orginates, and that the dispersion about the mean is fairly tight. The estimates for concentration parameters $a$ and $b$ of the beta variable $\varrho_u$, derived from $p$ and $\hat{\sigma}_p^2$, are respectively 3.86 and 4.50. $\hat{\kappa}_{SL}$ and $\hat{\kappa}_{nSL}$ are.08 and.79 respectively, signaling a much higher variance for the latent variable undergirding inter-county commuting volume than that of intra-county commuting.

## 5.1 Benchmarking with other delineations

In general, the clusters found by our proposed method are much larger than existing delineations and account for much more of the commuting activity. We define the *coverage rate* as the number of commuters between all edges of the given sets of clusters divided by the sum of

the total edges. This criterion captures how much of the total commuting activity is 'captured' by different modes of aggregation. The rate of coverage for all inter-county commuting was 86% for all clusters, compared to only 48% from OMB delineated MSAs and 77% from mega-regions. The coverage rate for all intra-county commuting was 92% from clusters imputed by CCME-SL, compared to 86% from MSAs and 74% from megaregions.

Similarities between clusters and MSAs are assessed using the Fuzzy Rand Index (Section 4.6) and by comparing size characteristics in Table 2. Clusters and MSAs are most alike in the West North Central and Mountain regions and least alike in the East South Central region. In the Mountain and Pacific regions, the number of clusters is nearly the same as the number of MSAs, and the mean commuter volumes captured by each method are more comparable than in the eastern states.

Clusters are most consistent with MSA delineations in large urban areas (see Figs 5 and 6). The size differences between the detected clusters and other delineations are muted in the regions around population centers such as Chicago and the cities of New York and Texas. The more populous and denser a *core* urban area is, the more concentrated a community (a cluster of more than one county) becomes. In Texas, for example, the economic regions surrounding major cities like Austin, Houston, Dallas, and San Antonio are all detected by our method, but these regions are larger and overlapping. The Austin region is the only Texas community that is drastically different from the corresponding MSA. The Houston region is split into coastal and inland commuting zones, which overlap at the core Houston area. The overlapping communities between Austin and San Antonio include Bastrop and Hays counties, identifying the dual memberships of these counties in larger cities' economic spheres of influence. Each of the four cities are also quantified as monads (not shown), signifying the relationship between large urban centers and their associated suburbs and hinterlands [16]. However, the larger, more diffuse nature of the communities signals emergent *lateral* commuting behavior between these peripheral counties [18].

Most urban-centric areas yield communities that are larger than and surround MSAs. One such example is the community surrounding Minneapolis, which is inclusive of the MSA but incorporates more peripheral counties (see Fig 5). In California, for example, a community in the Bay area region is similar to the San Francisco–Oakland–Fremont MSA, but the community also incorporates counties from San Jose–Sunnyvale–Santa Clara MSA (see Fig 4 because of the connected nature of inter-regional commuting in the Greater Bay Area. Northern California MSAs are similar to communities in the core area, but the community includes more peripheral counties.

**Table 2. Characteristics of clusters (monads, nodal and non-nodal communities) compared to MSA for each Census Division.**

| Census Division | FRI | # Clusters | | Mean Commuters | | Mean Size | | Coverage | | |
|---|---|---|---|---|---|---|---|---|---|---|
| | Clus/MSA | Clus | MSA | Clus | MSA | Clus | MSA | Cliques | Clus | MSA |
| Northeast | .66 | 8 | 15 | 703,315 | 133,223 | 10 | 2 | 86% | 92% | 60% |
| Midatlantic | .64 | 18 | 31 | 2,023,890 | 442,430 | 16 | 3 | 97% | 97% | 81% |
| E N Central | .63 | 24 | 70 | 1,557,092 | 179,574 | 22 | 2 | 81% | 83% | 68% |
| W N Central | .76 | 15 | 34 | 796,990 | 163,363 | 36 | 4 | 61% | 68% | 63% |
| S Atlantic | .60 | 33 | 80 | 1,225,996 | 195,596 | 32 | 4 | 85% | 90% | 69% |
| E S Central | .58 | 15 | 34 | 1,062,264 | 107,392 | 59 | 3 | 97% | 97% | 56% |
| W S Central | .66 | 33 | 43 | 661,861 | 230,651 | 19 | 3 | 76% | 85% | 70% |
| Mountain | .80 | 33 | 35 | 265,302 | 173,790 | 5 | 2 | 51% | 82% | 71% |
| Pacific | .70 | 42 | 43 | 706,892 | 321,860 | 5 | 2 | 47% | 79% | 74% |

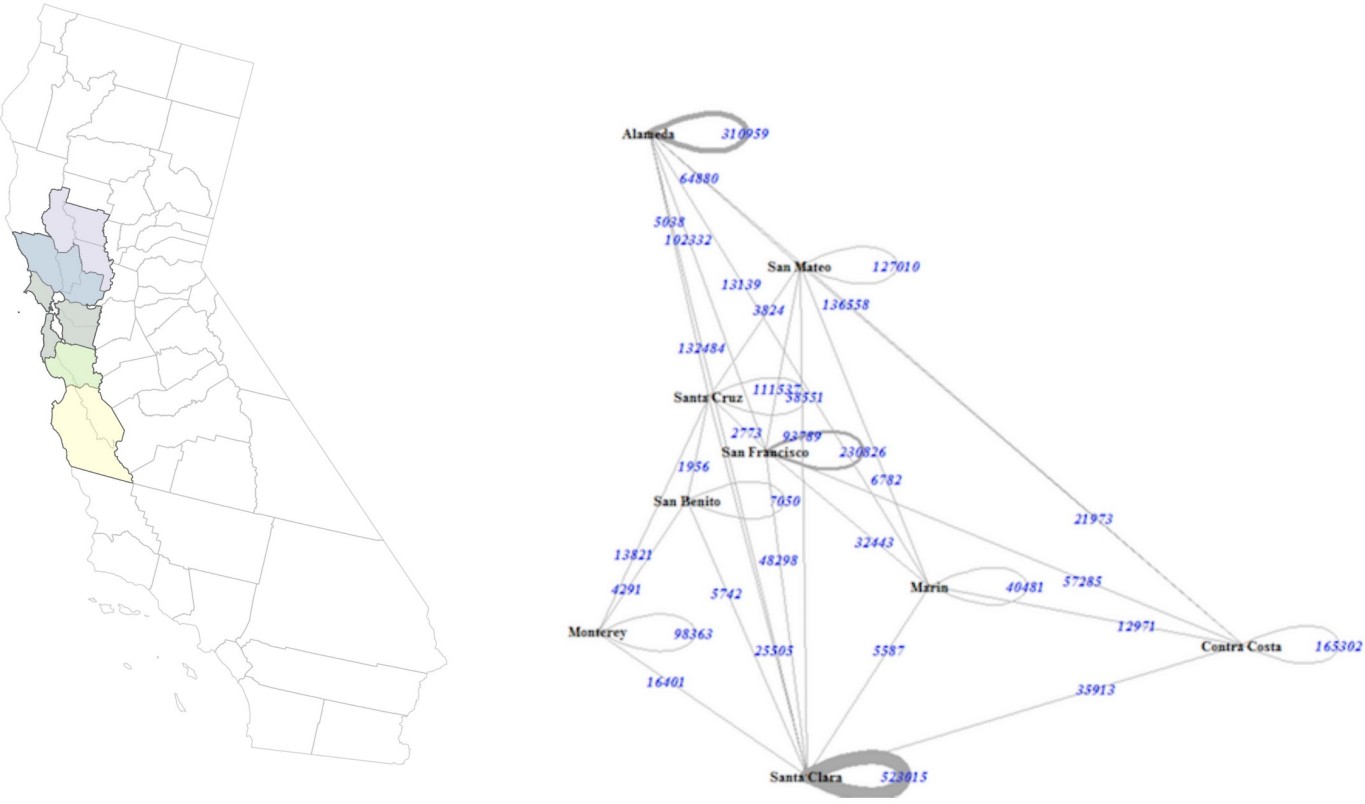

**Fig 4. Example of a tightly connected community in the Bay Area in Northern California.**

Some communities are similar in size but differ in shape from MSAs. Two of the communities associated with New York City are similar in size with their analogous MSAs, but communities do not incorporate as much of New Jersey counties as the MSA. One community in particular stretches North to upstate New York and part of New Hampshire.

Megaregions, in general, are larger than clusters from CCME-SL. The largest megaregion is in the Midwest connecting Minneapolis to Western Pennsylvania. The largest community yields a similar number of counties but is located in the Georgia-Tennesee region, centered around Atlanta (see Table 1). The largest communities appear to be in the southern central part of the US, where megaregions cover the least amount of ground (Fig 6). Though megaregions tend to be larger than the average community, the megaregions in the southern and central parts of the US still do not take into account the counties with smaller commuter volumes. The coverage rate for commuter volumes by megaregion is lower than that generated by communities for both intra-county and inter-county cases. The CCME-SL algorithm tends to find larger communities in counties that are more tightly connected by commuting, while megaregions tend to identify large population centers located in relative proximity to one another. The Piedmont megaregion stretches from Atlanta to central North Carolina and only comprises around 150 counties. Most communities around that region are focused on population centers. Cities like Tulsa in the central US are not accounted for by megaregions possibly because they do not have large population centers. However, these cities are still influential commuter hubs that facilitate the formation of large regions tied together by commuting.

The mean sizes of communities far exceed those of MSAs in all Census Divisions, except in the Western states, where the communities are only somewhat larger than MSAs (see Table 2). In the East South Central Division, the average cluster size is the largest at 59. In the Mountain and Pacific Divisions, however, the mean size of clusters is so small as to be almost on par with the number of MSAs. The largest delineated cluster is centered on the Atlanta area in the Southwest.

Communities (excluding monads) cover, on average, much more commuting volume than MSAs. In the Mountain and Pacific states, communities cover much less of the commuting volume because most populous counties are classified as monads. When taking into account monads, the coverage is higher than that of the MSAs in all regions. The proportion of commuting volume is nearly entirely captured by the clusters from CCME-SL in the East South Central and Midatlantic Divisons. 97% of total commuting activity is accounted for by clusters see Table 2). Communities and clusters (inclusive of monads) mostly cover above 80% of the commuting activity, while the MSAs cover less than 70%.

Many of the communities in the Midwest and South have numerous overlaps. For example, Cook and Tulsa counties are members of 5 different communities. North Carolina is covered by three large, heavily overlapping communities. Most nodal clusters are found in the Mountain West, where the counties are on average much larger and more sparsely settled. Monads are much more common in the Western US because the counties tend to be larger. Most major cities are identified as monads, including Cook, Harris, and Los Angeles. Notably, New York (Manhattan) and King (Brooklyn) counties are not monads because the counties that comprise New York City are all highly interconnected with each other (see Fig 5).

## 5.2 Comparison with other community detection methods

We compare the results found by CCME-SL with several other widely used methods. We examine the results imputed by modularity maximization (Louvain) and the degree-corrected stochastic blockmodel (DC-SBM). The Louvain method naturally finds the optimal number of partitions, while the DC-SBM needs a pre-specified number of partitions. In the Louvain, DC-SBM, and *expert judgment* (OMB) methods, regions are non-overlapping, while CCME-SL is the only method that demarcates in a way that allows counties to have multiple memberships (see Fig 6). A number of other techniques implicitly allow for self loops. CCME-SL was largely motivated by the need to address settings where self-loops account for a significant proportion of the weight emanating from a vertex.

The Louvain method imputes an optimal number of around 350 communities (see Fig 6, bottom left) and approximately maps commuting patterns to regions roughly similar in size to (large) CBSAs. We fit DC-SBMs under two different specifications for *number of blocks*: 100, which maps approximately to the number of clusters found by CCME-SL, and 350, which are the optimally clustered sets found by modularity maximization. We visualize these clusters alongside pre-defined MSA delineations and megaregions (see Fig 6).

Compared to DC-SBM and modularity methods, CCME-SL is capable of capturing overlapping communities finds much more variation in community sizes (though less so in weights). Counties that are influential for several regions, like Harris County in Texas, are strictly partitioned by DC-SBM and Louvain, but yield components in both 'coastal' and 'inland' counties in communities found by CCME-SL. Communities imputed by DC-SBM are highly dependent on the pre-specified number of blocks chosen. Los Angeles County is a single block under DC-SBM when 100 counties are chosen, but is subsumed by a much larger block when 350 blocks are chosen.

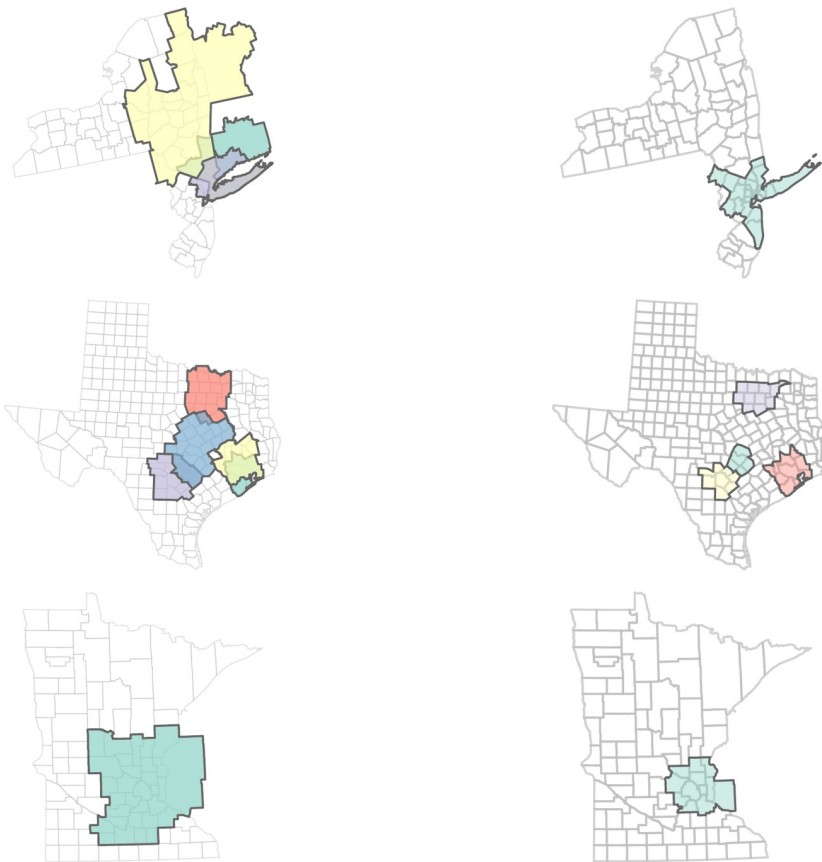

**Fig 5. Comparison of MSAs of New York City Region, major Texas cities, and Minneapolis(left) with their associated communities (right) in fairly populous regions.**

DC-SBM was implemented by means of regularized spherical spectral clustering [65], which has been shown to be consistent with DC-SBM in the package *randnet*. The modularity algorithm was implemented using the package *igraph*.

## 6 Discussion

We summarize our findings in this section and offer interpretations in relation to the economic geography of the US. We highlight how and why our findings are different from typical delineations and reconcile these findings with the introduced method and its novel incorporation of self-loops in a null model within a weighted network.

### 6.1 Substantive differences from existing agglomerations

Our method groups nearly all major urban areas in the U.S. into clusters. Counties like St Louis, Missouri and Fulton, Georgia (Atlanta) connect to many smaller, less populated counties. These clusters are much larger than those in more urban areas such as New York City or Harris (Houston). Counties in Southern California are not identified as multi-county communities but as monadic regions because these counties typically have much higher rates of intra-county commuting. However, in the Greater Bay area, Sacramento and Stockton are grouped

**CCME-SL**

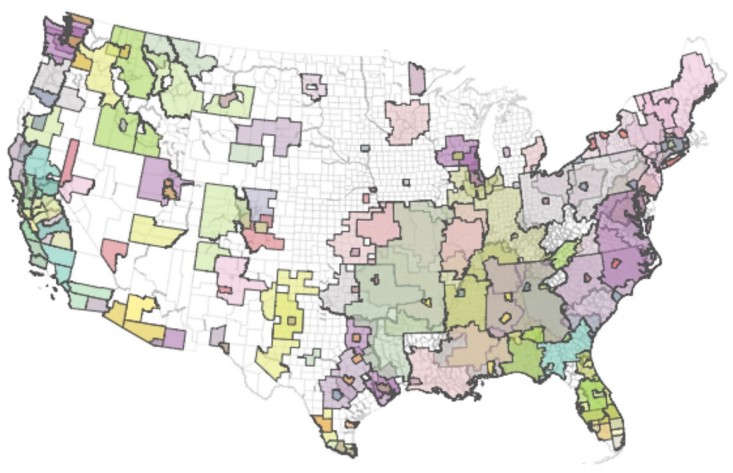

**Degree-Corrected Stochastic Blockmodel (100 Blocks (L), 350 Blocks (R))**

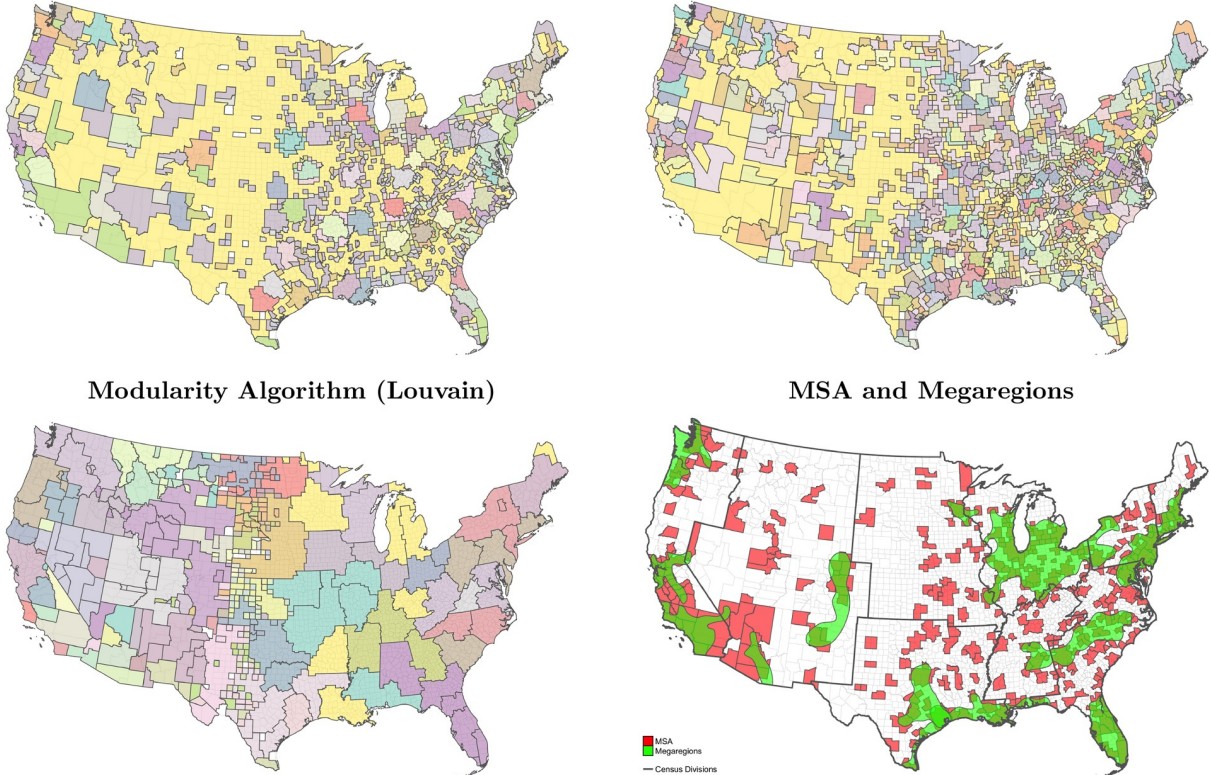

**Modularity Algorithm (Louvain)**          **MSA and Megaregions**

**Fig 6. Comparison of clusters from of CCME-SL (top) with DC-SBM (middle row, 100 blocks(left), 350 blocks (right)), Modularity (bottom left), and MSAs and megaregions (bottom right).**

together (see Fig 4). The clusters also overlap. Some counties in the Bay Area are also part of communities associated with the Sacramento metropolitan area.

In the Northeast, New York City is linked to upstate New York in some communities, Philadelphia is linked to Baltimore and Washington DC, and Baltimore/Washington DC are more linked to each other and to other urban centers along the Eastern seaboard stretching down to North Carolina. However, the Northeast and Midatlantic regions yield smaller communities on average compared to the South and Midwest. Because the commuting volumes in the Northeast are higher and counties are smaller, the counties in this region tend to cluster around each other in medium-sized communities of usually between 20-30 counties, which are larger than those in the West, but smaller compared to those in the South and Midwest.

Communities have some similarities to megaregions in the Midatlantic. New York, Philadelphia, Washington DC, and Baltimore are all grouped in the same megaregion, as is the case in several communities (see Fig 6). However, megaregions do not account for the connections between Baltimore, Washington DC and Eastern North Carolina. Communities generated by CCME-SL are allowed to overlap such that Eastern North Carolina and New York City are both *significantly* connected to Washington DC without being connected to each other. Such categorizations allow different urban areas to be flexibly classified in different regions.

The wide range of community sizes is due to both differing county sizes and differences in population between different regions of the United States. A county in California or Colorado tends to be at least a few times larger than a county in a state like Georgia. Thus, small communities are mostly found in the Western part of the contiguous US, while the largest clusters are found in the South and Central parts of the US.

Clusters from CCME-SL are larger than MSAs except in populous regions. CCME-SL naturally finds larger communities when inter-county commuting is relatively high compared to total commuting and smaller communities when large population centers are included. Communities show a large variation in size possibly because they account for the power-law distribution of human populations. Compared to MSAs, which are at most a few counties (excepting some metro regions like the Atlanta MSA), and megaregions, which are typically very large, community sizes are much more variable. That a single county can be compared to a massive region is a natural reflection of the highly uneven geographic distribution of the US population, and an expected feature of the results from an unsupervised method such as CCME-SL. Los Angeles County is the most populous county in the US with over 10 million inhabitants and 4 million commuters, a greater number of commuters than all counties combined in most states. LA County is identified as *monadic* and is within the same class of comparison with the 300-county communities stretching across several states in the South. Such a comparison does not make sense when only considering regions that are similar in size.

Communities found by the proposed method account for much more of the total inter-county commuting than is accounted for by MSAs. The discrepancy is especially large in the Southeast, Northeast, and Midwest, where communities and MSAs are most dissimilar according to FRI and differ most in the size, volume, and number of delineations (Table 2). MSAs in these regions capture between 50-70% of inter-county commuting compared to over 90% from communities. Counties in these regions are more interconnected by commuting than counties in other regions, making it likely that their corresponding MSAs are classifying counties in too fragmented a manner to effectively describe the highly interconnected nature of the regions (Table 2). In the Central and Southern US, interwoven commuting patterns are more spread out across counties, resulting in massive clusters.

CCME-SL produces clusters that vary greatly in size. The most populous MSAs house similar counties as their corresponding communities (see Fig 5). However, communities can also be as large as megaregions, although they tend to capture counties in different ways.

Megaregions capture cities that are in close proximity and which have large overall commuting volumes, but communities capture sets of counties that are closely linked by commuting, even when there are no cities and the gross commuting volume is not large. In the South and Central parts of the US, counties tend to be small and rural yet tightly interconnected. Such aggregations have not been depicted in existing official regional delineations and may be a novel contribution of the method in this study.

## 6.2 Methodological contributions to region demarcation

Though detection of monads may be antithetical to typical notions of *community* in network theory, they are very important in this particular application. This research highlights the role of self loops in commuting networks and is broadly applicable to human mobility networks with spatial constraints. As described in [66], most network data arising from nature adhere to power laws, and commuting flows are no exception. This study shows that power laws in spatial settings are intricately linked to self-referential behavior. Many studies have described human populations adhering to heavy-tailed distributions such as the Zipf Law [67, 68], but this study is among the first to indicate how regional delineations can account for such phenomena.

Strongly self-commuting counties that are identified as monads are also classified as clusters. Monads are found using tests of similar hypotheses evaluating how unlikely *the size of the total commuting activity in a given geographical unit is* compared to a scenario where commuting activity was generated at random subject to constraints arising from the weighted configuration model. The null hypothesis for monadicity parallels the null hypothesis for community connectivity in evaluating whether a test node is significantly more strongly connected to other nodes, or itself, than expected.

CCME-SL is mostly unsupervised. There are only two tuning parameters controlling the algorithm, $\alpha$ and $\tau$. $\alpha$ tunes the significance of the nodes' connectivity in relation to its associated community, and $\tau$ tunes the threshold of Jaccard distance to filter overlapping clusters. Different values of $\alpha$ and $\tau$ are used for both communities and monads, the results of which are shown in the Supporting Information. We set the $\alpha = 0.01$ and $\tau = 0.80$ for communities and $\alpha = 0.05$ for monads because monads cover nearly all the major US metropolitan areas. Though they don't differ much from those presented in Section 5, communities induced by different thresholds promote discovery of more tightly linked or more significantly *monadic* clusters.

Nelson and Rae use a multi-step routine to tune parameters for modularity-based methods, filtering out unwanted 'outlier' nodes and validating visual heuristics [22]. Compared to this approach, our method is more parsimonious and statistically interpretable.

## 6.3 Comparisons with other community detection methods

In Section 5.2, we compared the CCME-SL algorithm with several other standard community detection techniques, including the existing OMB demarcations. The primary advantage of CCME-SL is that it accounts for overlapping memberships for each node. Another advantage of CCME-SL is that the regions demarcated are more defensible because the model accounts for self-loops, which are important in the commuting network. Finally, the simplicity of CCME-SL is a practical advantage compared to other models, especially DC-SBM, which requires that the optimal clustering parameters be determined through cross-validation.

Though other techniques *implicitly account for* self-loops, the self-loops oftentimes cause distortions that create problems in identifying regions. Fig 6 shows that Louvain and DC-SBM yield smaller communities that are all roughly balanced in number of nodes per community.

Sizes of regions are highly contingent on their populations and commuting volumes. CCME-SL, on the other hand, yields clusters that are highly variable in size, commensurate with the highly variable populations. The sizes and characteristics of the clusters (monads and communities) imputed by CCME-SL thus appear to be *less constrained* than the other three methods shown in Fig 6. We posit that the differences between clusters from CCME-SL and other approaches are at least partly due to the extremely strong self-looping weights in high-strength nodes (populous counties).

Many algorithms return different results under different initialization scenarios when the likelihoods of partition functions are multimodal and thus give rise to a number of near-optimal partitions [69, 70]. Though this is a common problem in modularity-based approaches with random seedings, CCME-SL is not strongly affected by this issue for two different reasons. First, the method of initialization using the heuristics of starting at nodes with self-loops larger than 20,000 described in Section 4.1 produces the same results upon every run of the algorithm. Second, even if initialization was randomized and subject to different initialization criteria (while still retaining most of the population-center counties), the results do not look very different (see Fig 4 in S1 File). In other applications of CCME-SL, wherein the data do not have interpretable initial seeds, more runs would be required and analysis of *sets of partitions* would be necessary [35].

The number of communities is very important in community detection and its determination is a difficult problem in the field. For example, in DC-SBM the number can be determined by model selection or by cross validation. Under CCME-SL, the number of detected communities is determined by just one sample. This would be just one of the near-optimal states of the assumed model, as there would be other optimal states with different numbers of communities. Although the proposed method finds generally similar communities under a range of parameters and initializations (Figs 3,4 in S1 File), these validations do not allow for the discovery of some *exact* optimal objective. As such, we reiterate the point that CCME-SL should be viewed as an exploratory method that could give rise to more rigorous modes for proposing novel OMB-designated regions based on the structure of a commuting network.

## 6.4 Limitations and further research

Several limitations arise from the proposed method and application. Though mostly unsupervised, there are several tuning parameters that must be specified, such as the overlap parameter and the $\alpha$ threshold for the p-value. Sensitivity analysis (see Fig 3 in S1 File) over a range of p-values and overlap thresholds demonstrates that results do not change much when tuning parameters are tweaked. The post-processing step for determining the nodal communities does not capture the extent of the monocentricity that arises from areas that we expect to exhibit these properties, such as Multnomah County (Portland) and Hennepin County (Minneapolis).

Two major issues that we plan to explore in future work are:

1. **Resolution limit**: In future work we plan to explore the notion of resolution limit in the context of CCME and its judgement of significant communities. In brief, consider the simpler setting of unweighted networks. In an unweighted network the null model corresponding to an empirically observed network is created by placing an edge between two vertices $u$ and $v$ with a probability proportional to $d_u d_v / d_T$. For two sets $A$ and $B$ with degrees $d(A)$ and $d(B)$, $d(A) \times d(B) \ll d_T$ will cause the null model to find any presence of an edge between these sets to be "surprising" and cause $A$ and $B$ to be merged into the same community. This is a well-known issue with algorithms underpinned by a null model, as is the

case for modularity detection. Similar issues must also arise with CCME-like algorithms. One method to address this would be to incorporate a resolution parameter $\gamma$ as in the context of the unweighted case, which we now briefly describe (and refer the reader to work in [71] or [31] for more details). In the unweighted case, the natural null model is the configuration model which preserves degrees. This model implies that the null random graph model gives the probability of the existence of an edge between two vertices $u$ and $v$ as in (3). To accommodate for and deal with the issue of the resolution limit, the reference model can be modified so that

$$\mathbb{P}(\tilde{A}_{uv} = 1) = \gamma \frac{d_u d_v}{d_T}$$

As one increases $\gamma \uparrow \infty$, this implies that we expect non-trivial connectivity between nodes and thus connectivity within subsets to pass "higher bars" before being judged significant. We are exploring similar ideas in the context of the weighted case, including connections between modifications of the reference null model and the corresponding Markov stability of diffusion processes on the null models [72].

2. **Spatial null models**: In current work we are developing null models which also take into account the spatial component i.e. null models that directly include the spatial component and preserve various functionals such as the degree and the strength. We hope that this new approach will give more accurate results, mitigating issues like that of CCME-SL finding connections between neighboring counties surprising purely because of the resolution limit, since CCME-SL does not directly incorporate the notion that in spatial systems "neighboring counties have a higher propensity to connect".

Several fruitful directions of further research emerge from this study of commuting regions. This study documents the influence of self loops in spatial networks describing complex patterns arising from the collective behavior of many individuals. As network data in these realms become increasingly available, we expect the emergence of more methods accounting for self-looping behavior in networks. We plan to extend the methodology proposed in this study to directed networks so as to model the orientation of commuter flows. A further extension of the method described in this study is to use a temporal model to measure change in communities across time. Another avenue is to investigate the characteristics of power-law distribution of populations that are embedded within the commuting networks and how distributional characteristics of power laws interplay in community detection.

This research extends such a methodology to a more general setting of spatial networks that characterizes collective behaviors. Such networks often represent agglomerations of particles that are influenced by both core and peripheral elements. The method of community detection in networks with self-loops may be applied to a variety of spatially-constrained human mobility networks which naturally exhibit significant self-looping characteristics, such as human migration behaviors.

This method may find applications in other domains as well. In neuroimaging, one such application may be in analyzing the epicenter-spreading proliferation of biomarkers such as *tau*, which is significantly linked to Alzheimer's Disease. Research has revealed that tau develops along a trajectory of concentric spatial accumulation that aggregates at a seed region. Mapping such behavior in brain networks may find a suitable implementation in community detection on strongly self-looping graphs.

## 7 Conclusions

Traditional delineations of geographic regions have relied on agglomerations of smaller geographies, historical and political boundaries, separating edges and central foci. The boundary characterization is important not only for scientific purposes of tracking and tracing the historical evolution of urban systems, but also for administrative purposes of allocating infrastructure investments and formulating economic development strategies. Boundaries of metropolitan areas in the United States are artifacts of these delineation definitions, yet are central to tracking demographic and economic changes, funding allocations, determination of fair market rents, housing subsidies that depend on area median income and a host of other federal and state programs, even when the agencies caution their use for non-statistical purposes. These delineations are central but invisible to the lives of many. In this paper, we provide a robust method of accounting for the membership of a single place in multiple regions.

The main methodological contribution of this paper is its introduction of a community extraction method for a network with strongly self-looping characteristics. The application of the method on US commuting data suggests a way of conceiving delineations of economic geography that differs from existing approaches. CCME-SL accounts for intra-county commuting patterns and produces drastically different results when compared to other community detection methods as well as CBSA-based approaches. Furthermore, allowing regions to overlap allows us to create institutional structures and policies that are tailored not only to singular geographical entities, but also to multitudinous identities interacting across space and place.

## Appendix A. Probability Computations for Parameters

### A.1 Variance of $\xi_{uu}$: $\kappa_{SL}$

Note that

$$
\begin{aligned}
\mathrm{Var}(\xi_{uu}\varrho_u) &= \mathrm{Var}(\xi_{uu})\mathrm{Var}(\varrho_u) + \mathrm{Var}(\xi_{uu})(\mathbb{E}[\varrho_u])^2 + (\mathbb{E}[\xi_{uu}])^2\mathrm{Var}(\varrho_u) \\
&= \kappa_{SL}(\sigma_p^2 + p^2) + \sigma_p^2
\end{aligned}
$$

as the two variables are assumed to be independent. The sample variance of $W_{uu}$ may be decomposed in the following way:

$$
\begin{aligned}
\frac{1}{n}\sum_{u\in[n]}\mathrm{Var}(W_{uu}) &= \frac{1}{n}\sum_{u\in[n]}s_u^2\mathrm{Var}(\xi_{uu}\varrho_u) \\
&= \frac{1}{n}\sum_{u\in[n]}s_u^2\left(\kappa_{SL}(\sigma_p^2 + p^2) + \sigma_p^2\right) \\
&= \frac{1}{n}\sum_{u\in[n]}s_u^2\kappa_{SL}\left(\sigma_p^2 + p^2\right) + \frac{1}{n}\sum_{u\in[n]}s_u^2\sigma_p^2
\end{aligned}
$$

We derive another calculation of the sample standard deviation of self looping weights using a method of moments estimator.

$$
\frac{1}{n}\sum_{u\in[n]}\mathrm{Var}(W_{uu}) \approx \frac{1}{n}\sum_{u\in[n]}(W_{uu} - ps_u)^2
$$

From the above two equations, we derive the following approximation

$$\frac{1}{n}\sum_{u\in[n]}(W_{uu}-ps_u)^2 \approx \kappa_{SL}\frac{1}{n}\sum_{u\in[n]}s_u^2\left(\sigma_p^2+p^2\right)+\frac{1}{n}\sum_{u\in[n]}s_u^2\sigma_p^2$$

$$\Rightarrow \kappa_{SL}\sum_{u\in[n]}s_u^2(\sigma_p^2+p^2) \approx \sum_{u\in[n]}(W_{uu}-ps_u)^2-\sum_{u\in[n]}s_u^2\sigma_p^2$$

Rearranging the above equation and replacing unknown parameters by their estimates, we derive the estimate for $\hat{\kappa}_{SL}$

$$\hat{\kappa}_{SL}=\frac{\sum_{u\in[n]}(W_{uu}-ps_u)^2-\sum_{u\in[n]}s_u^2\sigma_p^2}{\sum_{u\in[n]}s_u^2(\hat{\sigma}_p^2+p^2)}$$

### A.2 Variance of $W_{uv}$

The properties of the weighted configuration model stipulate that the expectation of an edge weight given that there exist an edge, by Eq 5, is:

$$\mathbb{E}[W_{uv}|\mathbf{1}(A_{uv})]=(1-p)q_{uv}$$

$q_{uv}$ in the above equation is defined in the main body of the paper in Eq 6. We calculate the variance of $W_{uv}$ by decomposing it into two terms **I** and **II** using the following identity

$$\begin{aligned}\mathrm{Var}(W_{uv}) &=\mathbb{E}[\mathrm{Var}(W_{uv}|A_{uv})]+\mathrm{Var}(\mathbb{E}[W_{uv}|A_{uv}])\\ &=\mathbf{I}+\mathbf{II}\end{aligned} \quad (14)$$

To calculate **I**, we note that,

$$\mathrm{Var}(W_{uv}|A_{uv})=q_{uv}^2\mathrm{Var}((1-\varrho_u)\xi_{uv}|A_{uv})$$

Therefore, calculating **I** first necessitates calculating the conditional variance of $(1-\varrho_u)\xi_{uv}$, given that there exists an edge, under an expectation. As shorthand, we define the operation $\mathrm{Var}_A(\cdot):=\mathrm{Var}(\cdot|A_{uv})$ and $\mathbb{E}_A(\cdot):=\mathbb{E}_A[\cdot|A_{uv}]$.

$$\begin{aligned}\frac{1}{q_{uv}^2}\mathbb{E}[\mathrm{Var}(W_{uv}|A_{uv})\mathbf{1}(A_{uv})]=&\ \mathbb{E}[\mathrm{Var}((1-\varrho_u)\xi_{uv}|A_{uv})\mathbf{1}(A_{uv})]\\ =&\ \mathbb{E}[(\mathrm{Var}_A(1-\varrho_u)\mathrm{Var}_A(\xi_{uv})\\ &+\mathrm{Var}_A(1-\varrho_u)\mathbb{E}_A[\xi_{uv}]^2+\mathrm{Var}_A(1-\xi_{uv})\mathbb{E}_A[1-\varrho_u]^2)\mathbf{1}(A_{uv})]\\ =&\ \mathbb{E}[(\mathrm{Var}_A(\varrho_u)\kappa_{nSL}+\mathrm{Var}_A(\varrho_u)+\kappa_{nSL}(1-p)^2)\mathbf{1}(A_{uv})]\\ =&\ \mathbb{E}[(\sigma_p^2\kappa_{nSL}+\sigma_p^2+\kappa_{nSL}(1-p)^2)\mathbf{1}(A_{uv})]\\ =&\ \mathbb{E}[((\sigma_p^2+(1-p)^2)\kappa_{nSL}+\sigma_p^2)\mathbf{1}(A_{uv})]\end{aligned} \quad (15)$$

Since relation 15 holds for all $A_{uv}$, it becomes apparent that

$$\mathrm{Var}(W_{uv}|A_{uv})=q_{uv}^2((\sigma_p^2+(1-p)^2)\kappa_{nSL}+\sigma_p^2) \quad (16)$$

Using Eq 16, the calculation of **I** becomes straightforward:

$$
\begin{aligned}
\mathbf{I} &= \mathbb{E}[\mathrm{Var}(W_{uv}|A_{uv})] \\
&= q_{uv}^2 \mathbb{E}[(\sigma_p^2 + (1-p)^2)\kappa_{nSL} + \sigma_p^2)\mathbf{1}(A_{uv})] \\
&= q_{uv}^2 \cdot (\sigma_p^2 + (1-p)^2)\kappa_{nSL} + \sigma_p^2) \cdot \mathbb{P}(A_{uv}).
\end{aligned}
\tag{17}
$$

The calculation of **II**, similarly, is as follows:

$$
\begin{aligned}
\mathbf{II} &= \mathrm{Var}(\mathbb{E}[W_{uv}|A_{uv}]) \\
&= \mathrm{Var}(q_{uv}(1-p)\mathbf{1}(A_{uv})) \\
&= (1-p)^2 q_{uv}^2 \mathrm{Var}(\mathbf{1}(A_{uv})) \\
&= (1-p)^2 q_{uv}^2 \mathbb{P}(A_{uv})(1 - \mathbb{P}(A_{uv})).
\end{aligned}
\tag{18}
$$

Putting the two Eqs 17 and 18 together, we are able to solve for the variance of $W_{uv}$ from Eq 14, and substituting the expression for $\mathbb{P}(A_{uv})$ from Eq 3,

$$
\begin{aligned}
\mathrm{Var}(W_{uv}) &= \mathbb{E}[\mathrm{Var}(W_{uv}|A_{uv})] + \mathrm{Var}(\mathbb{E}[W_{uv}|A_{uv}]) \\
&= q_{uv}^2 \cdot (\sigma_p^2 + (1-p)^2) \cdot \kappa_{nSL} + \sigma_p^2) \cdot \mathbb{P}(A_{uv}) + q_{uv}^2 \mathbb{P}(A_{uv})(1 - \mathbb{P}(A_{uv}))) \\
&= q_{uv}^2 \mathbb{P}(A_{uv})((\sigma_p^2 + (1-p)^2) \cdot \kappa_{nSL} + \sigma_p^2 + 1 - \mathbb{P}(A_{uv})) \\
&= r_{uv}\left((\sigma_p^2 + (1-p)^2) \cdot \kappa_{nSL} + \sigma_p^2 + 1 - \frac{d_u d_v}{d_T}\right)
\end{aligned}
$$

where

$$
r_{uv} = \frac{\left(\frac{s_u s_v}{s_T}\right)^2}{\frac{d_u d_v}{d_T}} = q_{uv}^2 \mathbb{P}(A_{uv}).
\tag{19}
$$

### A.3 Variance of $\xi_{uv}$: $\kappa_{nSL}$

Now we construct a similar method of moments estimator for $\kappa_{nSL}$ as was done for $\kappa_{SL}$. However, we also make use of the expression for the conditional variance of $W_{uv}$ given the existence of an edge in Eq 16.

$$
\begin{aligned}
\sum_{u\in[n]}\sum_{v\neq u}\mathbb{E}[(W_{uv} - \mathbb{E}[W_{uv}])^2|A_{uv}] &\approx \sum_{u\in[n]}\sum_{v\neq u}\mathrm{Var}(W_{uv}|A_{uv}) \\
&= \sum_{u\in[n]}\sum_{v\neq u}((\sigma_p^2 + (1-p)^2)\kappa_{nSL} + \sigma_p^2 q_{uv}^2) \\
&= (\sigma_p^2 + (1-p)^2)\kappa_{nSL}\sum_{u\in[n]}\sum_{v\neq u}q_{uv}^2 + \sigma_p^2\sum_{u\in[n]}\sum_{v\neq u}q_{uv}^2
\end{aligned}
$$

After solving for $\kappa_{nSL}$ in the above equation and substituting unknown variables with their estimates, then changing the approximation to an equation, we derive the estimate $\hat{\kappa}_{nSL}$ for $\kappa_{nSL}$, thus obtaining the estimate as given in Eq 8:

$$
\hat{\kappa}_{nSL} = \frac{\sum_{u\in[n]}\sum_{v\neq u}(W_{uv} - (1-p)q_{uv})^2 - \hat{\sigma}_p^2\sum_{u\in[n]}\sum_{v\neq u}q_{uv}^2}{(\hat{\sigma}_p^2 + (1-p)^2)\sum_{u\in[n]}\sum_{v\neq u}q_{uv}^2}
$$

## A.4 Central Limit Theorem for $S(u, B, \mathcal{G})$ in set $B$

In this section we detail the calculation of the expectation and variance of the relative strength $S(u, B, \mathcal{G})$ of a given node-set $B$ used in iterative testing, described in the body of the text in Section 4.2.

$$S(u, B, \mathcal{G}) = \sum_{v \neq u, v \in B} (1 - \varrho_u) q_{uv} \xi_{uv}$$

$$= \sum_{v \neq u, v \in B} (1 - \varrho_u) \frac{\frac{s_u s_v}{s_T}}{\frac{d_u d_v}{d_T}} \xi_{uv}$$

Taking the expectation of each $\xi_{uv}$ gives the following expression for the strength of the node set

$$\mathbb{E}[S(u, B, \mathcal{G})] = (1 - p) \sum_{v \neq u, v \in B} \frac{d_u d_v}{d_T} \frac{\frac{s_u s_v}{s_T}}{\frac{d_u d_v}{d_T}}$$

$$= (1 - p) \sum_{v \neq u, v \in B} \frac{s_u s_v}{s_T}$$

$$= s_u \left( (1 - p) \sum_{v \neq u, v \in B} \frac{s_v}{s_T} \right)$$

We have found, in Section A.2, that the variance of a given $W_{uv}$ is expressed as:

$$\text{Var}(W_{uv}) = r_{uv} \left( (\sigma_p^2 + (1 - p)^2) \kappa_{nSL} + \sigma_p^2 + 1 - \frac{d_u d_v}{d_T} \right)$$

Adding the variance terms together in set $B$ yields:

$$\text{Var}(S(u, B, \mathcal{G})) = \sum_{u \neq v, u \in B} \text{Var}(W_{uv})$$

$$= \sum_{u \in B} r_{uv} \left( (\sigma_p^2 + (1 - p)^2) \kappa_{nSL} + \sigma_p^2 + 1 - \frac{d_u d_v}{d_T} \right)$$

Then given that $B$ is 'typical' and that $d_u$ and $B$ are sufficiently large, that $S(u, B, \mathcal{G})$ is approximately normal. For $\mu(u, B) = \mathbb{E}[S(u, B, \mathcal{G})]$ and $\sigma(u, B)^2 = \text{Var}(S(u, B, \mathcal{G}))$

$$\frac{S(u, B, \mathcal{G}) - \mu(u, B)}{\sigma(u, B)} \Rightarrow \mathcal{N}(0, 1) \tag{20}$$

Hence in each step of iterative testing in the update step of CCME, the normal p-value is used to iteratively reject insignificant nodes in a candidate community. Assumptions for and proofs of this Central Limit Theorem can be found in [56].

## Supporting information

**S1 Fig. Resulting communities from the CCME-SL algorithm.** Communities (non-nodal) (left), nodal clusters (middle), and monads (right).
(TIFF)

**S2 Fig. Example of a tightly connected community in the Bay Area in Northern California.**
(TIFF)

**S1 File.**
(ZIP)

## Acknowledgments

We thank two referees for an in depth reading of the entire manuscript and detailed comments that lead to a significant improvement of the original submission. We thank John Palowitch for helpful advice and expertise on the CCME method. We thank Professor Andrew Nobel for the initial development of the iterative testing methodology.

## Author Contributions

**Data curation:** Nikhil Kaza.

**Formal analysis:** Shankar Bhamidi, Nikhil Kaza.

**Investigation:** Shankar Bhamidi, Nikhil Kaza.

**Methodology:** Mark He, Joseph Glasser, Nathaniel Pritchard, Shankar Bhamidi, Nikhil Kaza.

**Project administration:** Shankar Bhamidi.

**Software:** Mark He, Joseph Glasser, Nathaniel Pritchard.

**Supervision:** Shankar Bhamidi.

**Visualization:** Nikhil Kaza.

**Writing – original draft:** Mark He, Shankar Bhamidi, Nikhil Kaza.

**Writing – review & editing:** Mark He, Joseph Glasser, Shankar Bhamidi, Nikhil Kaza.

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
