## [Decision Letter · Decision Letter 0]

26 Jul 2019

PONE-D-19-14845

Demarcating Geographic Regions using Community Detection in Commuting Networks

PLOS ONE

Dear Dr. Kaza,

Thank you for submitting your manuscript to PLOS ONE. After careful consideration, we feel that it has merit but does not fully meet PLOS ONE’s publication criteria as it currently stands. Therefore, we invite you to submit a revised version of the manuscript that addresses the points raised during the review process.

We would appreciate receiving your revised manuscript by Sep 09 2019 11:59PM. To enhance the reproducibility of your results, we recommend that if applicable you deposit your laboratory protocols in protocols.io, where a protocol can be assigned its own identifier (DOI) such that it can be cited independently in the future. For instructions see: http://journals.plos.org/plosone/s/submission-guidelines#loc-laboratory-protocols

We look forward to receiving your revised manuscript.

Kind regards,

Renaud Lambiotte

Academic Editor

PLOS ONE

Journal Requirements:

3. In your data availability statement you write, "All relevant data are within the paper and its Supporting Information files." Please ensure you have listed the third party data sources for your study and information on how other researchers could access that data (http://journals.plos.org/plosone/s/data-availability#loc-faqs-for-data-policy). If you have the necessary permissions, please consider uploading the data used in your specific study to a data repository such as Dryad or Figshare.

Reviewers' comments:

Reviewer's Responses to Questions

**Comments to the Author**

1. Is the manuscript technically sound, and do the data support the conclusions?

Reviewer #1: Partly

Reviewer #2: Partly

2. Has the statistical analysis been performed appropriately and rigorously? 

Reviewer #1: No

Reviewer #2: Yes

3. Have the authors made all data underlying the findings in their manuscript fully available?

Reviewer #1: Yes

Reviewer #2: No

4. Is the manuscript presented in an intelligible fashion and written in standard English?

Reviewer #1: Yes

Reviewer #2: No

5. Review Comments to the Author

Reviewer #1: The authors identified the clusters of counties in US from commuting data among them, using by a overlapping community detection method with self-loops.

Such a data-driven approach to identify spatial units would be promising, overcoming the problems of the traditional heuristic methods. On the other hand, this is not the first paper for this approach, and the reliability of their method should be checked carefully to use the results for statistical, governance and planning purposes that the authors argued in the paper.

1.

There are some papers to identify spatial units from the human mobility datasets.

For examples,

- The impact of social segregation on human mobility in developing and industrialized regions by Amini, A., Kung, K., Kang, C. et al. EPJ Data Sci. (2014) 3: 6. https://doi.org/10.1140/epjds31.

- Comparing Community Detection Algorithms in Transport Networks via Points of Interest by Huang, Liping & Yang, Yongjian & Gao, Hepeng & Zhao, Xuehua & Du, Zhanwei. (2018). IEEE Access. PP. 1-1. 10.1109/ACCESS.2018.2841321.

- Detecting the dynamics of urban structure through spatial network analysis by Chen Zhong and Stefan Müller Arisona and Xianfeng Huang and Michael Batty and Gerhard Schmitt, International Journal of Geographical Information Science

Volume 28, 2014 - Issue 11. https://doi.org/10.1080/13658816.2014.914521

- The Size Distribution of 'Cities' Delineated with a Network Theory-based Method and Smartphone GPS Data by Shota Fujishima, Naoya Fujiwara, Yuki Akiyama, Ryosuke Shibasaki, and Hodaka Kaneda, Center for Spatial Information Science, Discussion Paper http://www.csis.u-tokyo.ac.jp/wp-content/uploads/2018/02/150.pdf

This is not a comprehensive list of the topic(I just searched in a web).

Please add a short review in Introduction and give sufficient credits to such previous works.

I also think this paper contains original results to be published, because of the new method of community detection with the application to another dataset.

2.

Concerning the community detection method, I think the reliability is not checked enough to use it for the purposes addressed in this paper.

The authors aimed to identify spatial areas, instead of CBSA(core based statistical areas) or MSA(metropolitan areas), for examples.

Such areas are the basis of many statistics in US, and should be carefully considered to guarantee the reliability of the detection method.

The community detection methods, however, have many problems in general.

This is mainly because the likelihood function is multimodal and there are a number of near-optimal partitions, as discussed in literature.

- B.H. Good, Y.-A. de Montjoye and A. Clauset, "The performance of modularity maximization in practical contexts." Physical Review E 81, 046106 (2010).

- Peel, Leto & B. Larremore, Daniel & Clauset, Aaron. (2016). The ground truth about metadata and community detection in networks. Science Advances. 3. 10.1126/sciadv.1602548.

This would be problematic to obtain reliable results.

I suspect that the author's optimization method focuses only a single partition that would be optimal, and can not deal with the problem property.

Recently, network scientists have examined the ensembles of partitions, not just focusing on the optimal one. For a example, the number of clusters are determined by the model selection or model evidence in Bayesian inference framework.

- A review, Tiago P. Peixoto, “Bayesian stochastic blockmodeling”, arXiv: 1705.10225.

It would be better if the authors carefully consider this issue and develop their method.

At least, the authors should address the issue in Discussion and argue on the potential limitations of their methods.

3.

The proposed method is not clear, because

(i) Two different parts of the method are mixed. The theoretical description of the statistical model and its numerical implementation should be described separately.

(ii) The relation to the previous models is unclear.

Related to the second point, the proposed method seems to be non-standard in network science.

This is okay, but I would like to ask about the resolution limit, that is already studied in the cases of standard methods.

This resolution limit is significantly related to the issue of the detection of `Monads', obtained by self-loop consideration.

Is the limit is small enough to detect a single node as a community?

Moreover, the authors are responsible to allocate their method in literature to join the scientific discussions. The current manuscript is not enough for readers to understand the relationships to previous methods, related to the point (i).

For example, could you formulate the model in the Bayesian framework and argue the differences from degree-corrected SBM?

4.

The details of the proposed model seems to be unnatural to model the commuting data.

In the null model, adjacency matrix A and weight matrix W is separately used.

(After assigning edges, the weight is allocated only to existing edges.)

But, the commuting data is represented only by W, and A is just its binarization by a threshold of 100 commuters.

Why not consider a model of W, without A?

What is the aim to introduce A?

Please explain the logic behind the modelling.

In addition, the threshold seems to be ad-hoc.

5.

Authors argued the novelty of introduction of self-loops to community detection methods.

But, I think, this is not novel. The previous studies have also considered it, and in most cases, there is no limitation on self-loops, except that the methods are based on tree-like approximations in sparse graphs.

It just depends on datasets in practice.

I agree that the self-loops will be important in the commuting data studied in this paper.

But, the self-loops will not so important in higher-resolution mobility data, such as 500mx500m grid.

Today, because of GPS and other technologies, we are getting such nice datasets, and in the cases, the commuters within the same county are recorded as those between different locations and self-loops will be rare.

Please rewrite your arguments on the novelty considering these points.

6.

I would like to see the result of the community detection in entire US, and visually compare it with the other partitions of MBA, megaregions, and other partitions of standard methods (if possible. e.g. degree-corrected SBM and modularity).

Then, the results of the paper would be more effectively presented, and facilitate further scientific discussions.

It is possible plot the overlapping communities using pi charts.

https://graph-tool.skewed.de/static/doc/_images/lesmis-sbm-marginals.svg

Reviewer #2: Q1: I think that the method is technically sound from the point of view of the mathematics in it, but I do not think the data support the conclusions. I disagree that including self-edges is novel in the field of community detection (although towards the end in Section 6.4 the authors explain that they mean the use of community detection to study commuting networks). I also believe that one of the biggest points that the work argues is that the proposed method has a high coverage of the commuter travels, but I fail to see why that is the determinant justification of the method working. For example, the coverage can be trivially maximised by taking a single community that covers all of the US, but such a community is obviously not relevant. This fits into the larger issue of validation, of how can the communities found be interpreted (a general problem in the field of community detection) and I think that finding statistically significant will only give you confidence with respect to the thing you are trying to explain (coverage as compared to the null model). I am sceptical that some of the largest communities (such as the light blue in fig 2b) are useful. Finally, I think that the claims made in the conclusion about dramatically changing our understanding of the metropolitan structure of the US are not supported by the work presented before.

Q2: I think that the statistical analysis was done rigorously. However, something that is not entirely clear to me from the text is if the CCME algorithm is proposed here or if it has been proposed somewhere else (in which case a citation would be needed). It's not entirely clear why the calculation of the variances is relevant for the analysis and it should be explained in the text (and I am not recommending to remove the calculations). In section 7 the text says that the sensitivity analysis can be found in the supplementary material and this is not the case.

Q3: As far as the material that was available to download as an editor, I did not have access to any type of data. This includes data of commuters, networks build, maps, figures and anything that would be needed to reproduce the analysis.

Q4: The article is written in generally good English but it has not been revised with care. There are many instances of missing words or extra words, specially towards the end of the article. More citations should be added, for example for the Benjami-Hochberg procedure. One further comment is that the terminology that is new to the article should be presented at the beginning, not in Section 5 (Results) as is the case for non-nodal community, monad, etc. While I agree that it is necessary to distinguish the different types of communties found, I think that in the context of the field is is confusing to use the work cluster to mean a community that contains more than one node. To the best of my knoledge, the word monad has been introduced in this manuscript and this is also a non-standard term. I would prefer the self-explanatory single-node community.

6. PLOS authors have the option to publish the peer review history of their article (what does this mean?). If published, this will include your full peer review and any attached files.

Reviewer #1: No

Reviewer #2: No

---

## [Author Response · Author response to Decision Letter 0]

3 Oct 2019

A separate document that individually addresses the reviewers' concerns is attached to the submission.

---

## [Decision Letter · Decision Letter 1]

20 Dec 2019

PONE-D-19-14845R1

Demarcating Geographic Regions using Community Detection in Commuting Networks

PLOS ONE

Dear Dr. Kaza,

Thank you for submitting your manuscript to PLOS ONE. After careful consideration, we feel that it has merit but does not fully meet PLOS ONE’s publication criteria as it currently stands. Therefore, we invite you to submit a revised version of the manuscript that addresses the points raised during the review process.

We would appreciate receiving your revised manuscript by Feb 03 2020 11:59PM. To enhance the reproducibility of your results, we recommend that if applicable you deposit your laboratory protocols in protocols.io, where a protocol can be assigned its own identifier (DOI) such that it can be cited independently in the future. For instructions see: http://journals.plos.org/plosone/s/submission-guidelines#loc-laboratory-protocols

We look forward to receiving your revised manuscript.

Kind regards,

Renaud Lambiotte

Academic Editor

PLOS ONE

Reviewers' comments:

Reviewer's Responses to Questions

**Comments to the Author**

1. If the authors have adequately addressed your comments raised in a previous round of review and you feel that this manuscript is now acceptable for publication, you may indicate that here to bypass the “Comments to the Author” section, enter your conflict of interest statement in the “Confidential to Editor” section, and submit your "Accept" recommendation.

Reviewer #1: (No Response)

Reviewer #2: (No Response)

2. Is the manuscript technically sound, and do the data support the conclusions?

Reviewer #1: Yes

Reviewer #2: Yes

3. Has the statistical analysis been performed appropriately and rigorously? 

Reviewer #1: Yes

Reviewer #2: Yes

4. Have the authors made all data underlying the findings in their manuscript fully available?

Reviewer #1: Yes

Reviewer #2: Yes

5. Is the manuscript presented in an intelligible fashion and written in standard English?

Reviewer #1: Yes

Reviewer #2: No

6. Review Comments to the Author

Reviewer #1: The revised manuscript has addressed the issues I pointed out, and the authors have responded them carefully. I still have some different opinions on the issues, but I also think that such differences are not critical and should be judged by readers.

However, there is one exception related to the point (3). The authors argued that the output of the method is stable if the initial condition is fixed, and the outputs are robust against the change of the initial conditions. This is necessary requirement, but, it is not enough.

To clarify the addressed issue, I'd like to focus on a single aspect of the output of the method. This is the number of the detected community by their method. This number is very important and should be determined on a reasonable basis. For example, in DC-SBM, the number can be determined by model selection or by cross validation (in this paper the number is pre-determined by authors in sec. 6. This should be updated, too). On the other hand, in the proposed method, the number of detected communities is determined by just one sample. This would be one of near optimal states of the assumed model, but there would a lot of another optimal states with different numbers of communities. Thus, there is no reasonable validation whether the detected number is good or not.

I think this is a difficult problem, and my question did not mean that this issue should be solved perfectly. I also think that the proposed method is good enough to present a new visualization tool of geographic regions. But, this point can be problematic if this paper is still aimed to propose a new identification method for grouping counties for further statistics, and such limitations should be noted.

Reviewer #2: I thank the authors for addressing most of my concerns and I think that the manuscript is much improved. I have a few final comments.

1. I did not mean that monad should be changed for 'self-explanatory single-node community' which I agree is a mouthful. I was suggesting 'single-node community' and I meant that this term was self-explanatory. I still have the sense that introducing so much new terminology as the authors do makes is less intelligible to their peers. However, this is just a suggestion and I leave it now to their best judgement after having explained myself.

2. While the manuscript is much improved in the explanation of the terms, there are still a couple of things that are confusing. In the abstract the term 'non-trivial' to refer to self-loops is really confusing. What is a trivial self-loop? We have to wait until page 12 to find out, which is very bad practice for an abstract. Idem for the term propensity. It is defined in page 6 but discussed much before then. This is also not a standard term, and I would suggest removing it entirely and explaining instead that you mean the proportion of the inter-county commuting flow compared to the total outward flow.

3. The authors discuss the other community detection methods in terms of a comparison with a null model. They imply that this is done within the SBM framework, but this is not the case. In this framework we decide to model the data with a particular type of model, and then we fit the model parameters. In page 5 they talc about the 'structured SBM' and I don't know what structured means here.

4. I would swap the sections 4.1 and 4.2 because the notation comes before it is needed, and more discussion follows.

5. A minor point, but I see no technical reason why the degree of node u needs to exclude self-loops (the degree is fact is almost not used). To make this more similar to the strength which does include node u, it is convenient to define A_uu = 2 as a convention, and in this case the normal degree formulas work and the degree of each node (with self-loops) simply increases by 2.

6. While some changes have been made and the authors acknowledge that the SBM methodology accomodates for self-edges, the text still says that most other methods and in particular modularity in general does not account for self-loops. This is not the case, and in fact the Newman-Girvan null model implicitly allows self-loops.

7. In page 10 there are two instance of \\hat{\\rho}_u that seem to be referring to two different quantities, an empirical and a simulated one.

8. Figure 6 should be moved before the references. This figure really adds to the manuscripts and in particular it is easy to see that the other methods find a very different structure (for example the 'background noise' that their method disregards). I would point out in the text in section 6.2 that the methods give non-overlapping communities (i.e. partitions as opposed to covers) (althought the SBM can be used to get covers, but this is probably much more work and not necessary).

9. I did not understand the point of section 7.4, which is completely out of place with the discussion before and after (notably with the definition of modularity).

10. Final comment, the text still needs more a good proofread. There a new errors (still missing or extra words), and in the new section 2.2 the authors cannot use '[2]' as a subject, but they can write 'In reference '[2]', the authors do X and Y'. The numbers cannot be used as a subject in the sentence.

7. PLOS authors have the option to publish the peer review history of their article (what does this mean?). If published, this will include your full peer review and any attached files.

Reviewer #1: No

Reviewer #2: No

---

## [Author Response · Author response to Decision Letter 1]

4 Feb 2020

Please refer to the Response to the Reviewers document.

---

## [Editor Report · Decision Letter 2]

13 Mar 2020

Demarcating Geographic Regions using Community Detection in Commuting Networks with Significant Self-Loops

PONE-D-19-14845R2

Dear Dr. Kaza,

We are pleased to inform you that your manuscript has been judged scientifically suitable for publication and will be formally accepted for publication once it complies with all outstanding technical requirements.

With kind regards,

Akbar Ali

Academic Editor

PLOS ONE
---

## [Editor Report · Acceptance letter]

3 Apr 2020

PONE-D-19-14845R2 

Demarcating Geographic Regions using Community Detection in Commuting Networks with Significant Self-Loops 

Dear Dr. Kaza:

I am pleased to inform you that your manuscript has been deemed suitable for publication in PLOS ONE. Congratulations! Your manuscript is now with our production department. 

With kind regards,

on behalf of

Dr. Akbar Ali 

Academic Editor

PLOS ONE